# Divergent venom effectors correlate with ecological niche in neuropteran predators
Maike Laura Fischer[1], Henrike Schmidtberg[2], Olivia Tidswell[1], Benjamin Weiss[1], Ludwig Dersch[3,4], Tim Lüddecke[3,4], Natalie Wielsch[5], Martin Kaltenpoth [1], Andreas Vilcinskas[2,3,4] & Heiko Vogel [1] ✉

Neuropteran larvae are fierce predators that use venom to attack and feed on arthropod prey. Neuropterans have adapted to diverse and sometimes extreme habitats, suggesting their venom may have evolved accordingly, but the ecology and evolution of venom deployment in different families is poorly understood. We applied spatial transcriptomics, proteomics, morphological analysis, and bioassays to investigate the venom systems in the antlion *Euroleon nostras* and the lacewing *Chrysoperla carnea*, which occupy distinct niches. Although the venom system morphology was similar in both species, we observed remarkable differences at the molecular level. *E. nostras* produces particularly complex venom secreted from three different glands, indicating functional compartmentalization. Furthermore, *E. nostras* venom and digestive tissues were devoid of bacteria, strongly suggesting that all venom proteins are of insect origin rather than the products of bacterial symbionts. We identified several toxins exclusive to *E. nostras* venom, including phospholipase A2 and several undescribed proteins with no homologs in the *C. carnea* genome. The compositional differences have significant ecological implications because only antlion venom conferred insecticidal activity, indicating its use for the immobilization of large prey. Our results indicate that molecular venom evolution plays a role in the adaptation of antlions to their unique ecological niche.

The Neuroptera are a diverse order of insects comprising more than 6500 holometabolous species assigned to 16 extant families, including the lacewings (Chrysopidae) and highly specialized antlions (Myrmeleontidae)[1,2]. Neuropterans diverged from the remaining Neuropterida during the Permian period and were one of the first groups of insects to evolve venom as a predation mechanism[1–3]. In contrast to many other venomous insects, only the larval stages produce venom. Their specialized piercing-sucking mouthparts form elongated pincers, which are used to catch and feed on prey using extraoral digestion (EOD)[4–6]. The appearance, ecology, and physiology of neuropteran larvae are diverse and differ greatly from other insect orders[1,2,7,8]. Moreover, neuropterans have adapted to a wide range of diverse and in some cases extreme habitats[7,9–11].

The best-studied neuropteran family (Chrysopidae) includes the green lacewing *Chrysoperla carnea* (Stephens, 1836), a beneficial insect used in pest control[12,13]. *C. carnea* larvae, also known as aphid lions, are active predators that feed on other small, soft-bodied arthropods, and are well known for their ability to consume large numbers of aphids[14,15]. Their short development time and low burden on their habitat provide ideal conditions

for mass rearing[16–19]. In contrast, antlion larvae are sit-and-wait predators that have adapted to dry and sandy habitats[9,20,21]. They typically have a long larval development phase and can survive extensive periods of starvation, a critical adaptation that accommodates fluctuating prey availability in harsh environments[22–24]. Some species, such as the European antlion *Euroleon nostras* (Geoffroy, 1785), build pitfall traps in the sand that provide shelter and protection, but primarily facilitate prey capture by causing small animals to fall into the traps[9,25]. Antlions like *E. nostras* combine such traps with potent, paralyzing venom to overwhelm relatively large prey[5,26,27].

The ecology and evolution of venom use in differently-adapted neuropteran families is not well understood. In particular, antlions appear to secrete potent insecticidal venom but not much is known about its composition, mode of action, and role in their extreme lifestyle[26,28]. Several studies have reported that bacterial symbionts facilitate toxin production in the antlion *Myrmeleon bore* (Tjeder, 1941) and identified putative venom toxins of bacterial origin[29–32]. However, there is a lack of molecular data to confirm the presence and ecological relevance of bacterial toxins in neuropteran venoms. Moreover, the glandular origin of neuropteran venom is

[1]Department of Insect Symbiosis, Max Planck Institute for Chemical Ecology, Jena, Germany. [2]Institute for Insect Biotechnology, Justus Liebig University, Giessen, Germany. [3]Branch Bioresources of the Fraunhofer Institute for Molecular Biology and Applied Ecology, Giessen, Germany. [4]LOEWE Centre for Translational Biodiversity Genomics (LOEWE-TBG), Frankfurt, Germany. [5]Research Group Mass Spectrometry/Proteomics, Max Planck Institute for Chemical Ecology, Jena, Germany. ✉e-mail: hvogel@ice.mpg.de

still unclear. Most mouthpart-associated venom systems in insects evolved from salivary glands, but the morphology and homology of the neuropteran venom system are not fully understood[3,33]. Some studies suggest that neuropterans inject regurgitant from the gut[5,26], whereas others have identified two glandular structures at the base of the maxillae – usually referred to as venom gland and lateral gland, respectively – and a maxillary-mandibular gland, all of which may contribute to venom production[33–36].

We investigated the tissue origin, composition, and ecological role of antlion venom in *E. nostras* using a combination of (spatial) transcriptomics, proteomics, and morphological analysis, the latter consisting of fluorescence in situ hybridization (FISH), micro-computed tomography (μCT), hybridization chain reaction (HCR)-RNA-FISH and standard

histology (Fig. 1). We characterized the structures involved in venom production and storage, addressed the potential contribution of microbial symbionts and compared protein composition and toxicity of *E. nostras* and *C. carnea* venom. To evaluate the role of venom evolution in the adaptation of antlions to their extreme habitats, we also identified genes encoding for venom effector proteins exclusively found in the antlion genome.

## Results

### Morphology of the neuropteran venom system

We reconstructed the digestive tract and putative venom system of *E. nostras* and *C. carnea* using a combination of μCT and histological analysis. The general morphology was similar in both species. The food canal, formed

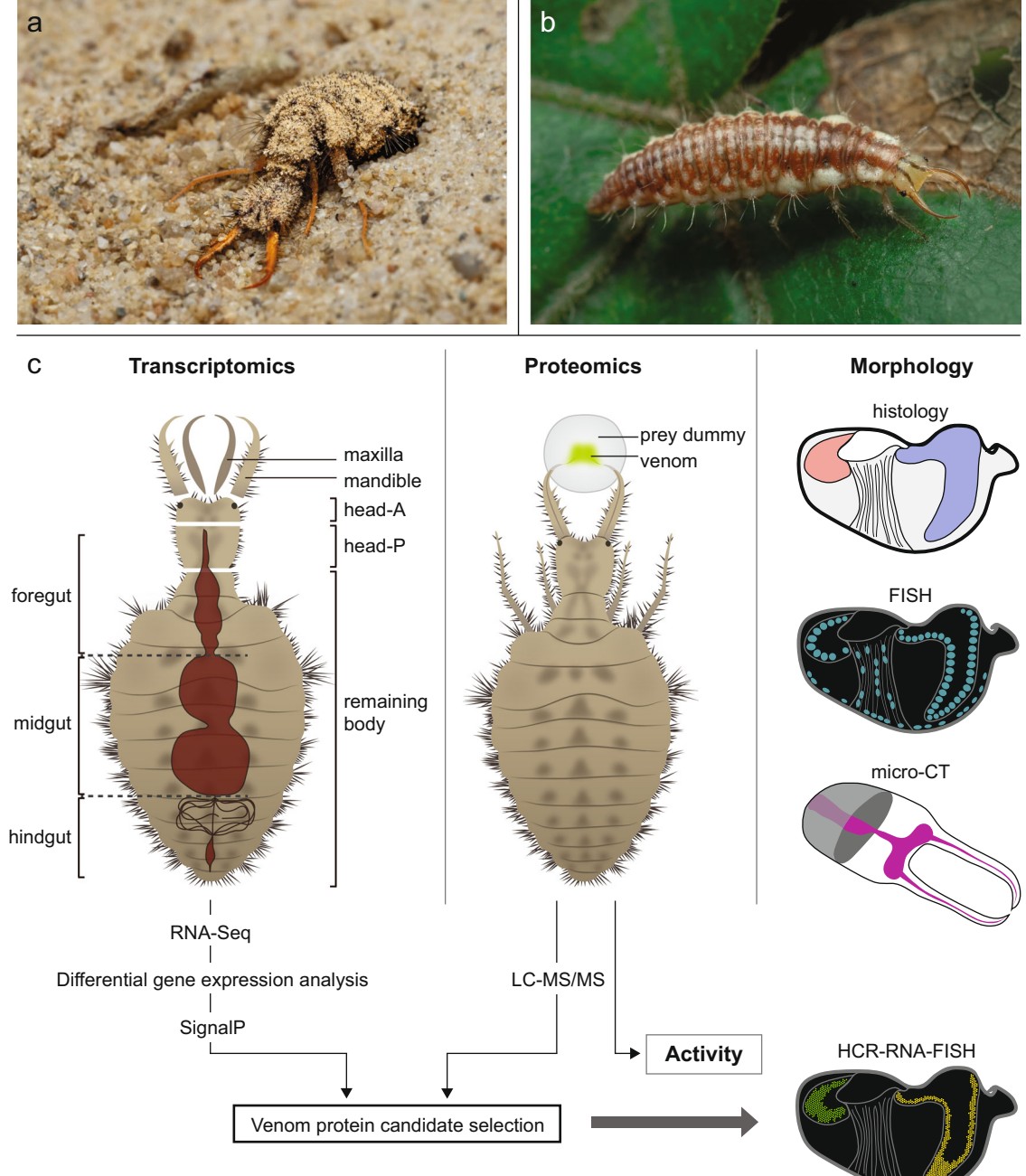

**Fig. 1 | Overview of the study organisms and methods. a** *Euroleon nostras* larva in its natural habitat. Photo: Benjamin Weiss. **b** *Chrysoperla carnea* larva in its natural habitat. Photo: Devasena Thiagarajan. **c** Schematic overview of the experimental approach. We combined transcriptomic, proteomic, and morphological analysis,

including histology, fluorescence in situ hybridization (FISH), micro-computed tomography (μCT), and hybridization chain reaction (HCR)-RNA-FISH) to describe the glandular origin, composition, and ecological role of venom in the European antlion *E. nostras* and the green lacewing *C. carnea*.

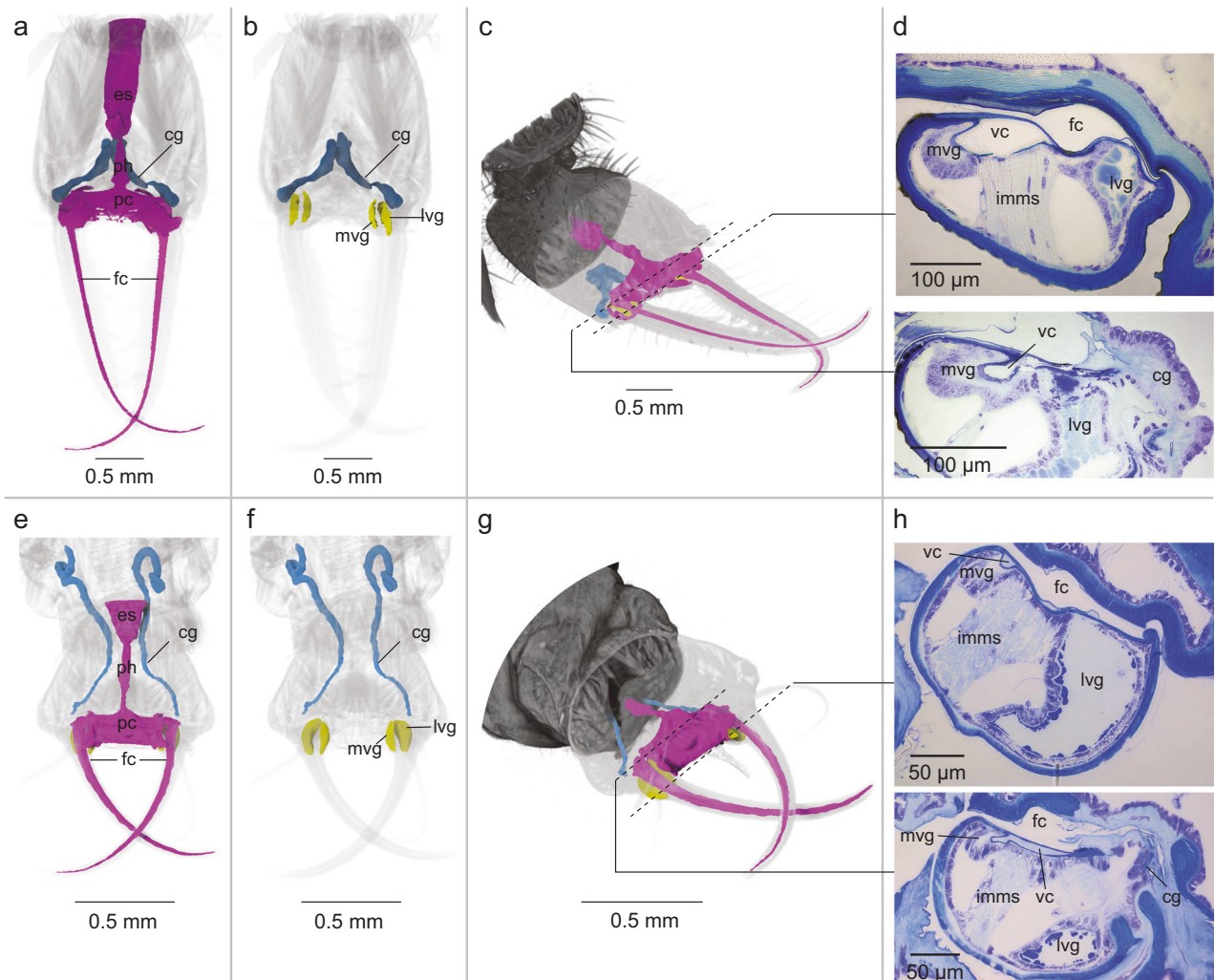

**Fig. 2 | Micro-CT and histological analysis of the neuropteran venom system and digestive tract. a–d** Reconstructions of the venom glands and digestive tract of *Euroleon nostras*. **e–h** Reconstructions of the venom glands and digestive tract of *Chrysoperla carnea*. In both species, the food canal (fc) leads from the tips of the mouthparts into the head and opens into a branched pre-oral chamber (pc) that narrows posteriorly to form the pharynx (ph) and esophagus (es). The medial venom gland (mvg) and lateral venom gland (lvg) are surrounded by the pre-oral chamber at the base of the maxillae and are separated by the intrinsic muscle of the maxillary stylet (imms). The cephalic gland (cg) opens at the base of the food canal and extends posteriorly into the head.

by the interlocking mandibles and maxillae, leads from the tips into the head and opens into a branched pre-oral chamber that narrows posteriorly to form the pharynx and esophagus (Fig. 2a, c, e, g). A venom canal lies inside the maxillae, parallel to the food canal but spatially separated from it (Fig. 2d, h). At the base of each maxilla, separated by the intrinsic muscle of the maxillary stylet and connected to the base of the corresponding venom canal, are two distinct glandular structures. These maxillary glands are located within the maxillae on either side of the venom canal and form the organs that have been described as the venom gland and lateral gland, but are referred to here as the medial venom gland and lateral venom gland, respectively (Fig. 2b, d, f, h). A cephalic gland (formerly also described as mandibular-maxillary gland) opens at the base of the food canal (Fig. 2f, h) and extends posteriorly into the head (Fig. 2b, d). Additional µCT and histological images are provided in Figure S1 and S2, respectively.

### Protein composition of neuropteran venoms

Proteotranscriptomic analysis of venom collected using a prey dummy (Fig. 1) was carried out by tissue-dependent RNA-Seq (Table S1) and liquid chromatography-tandem mass spectrometry (LC-MS/MS). This revealed that both *E. nostras* and *C. carnea* larvae inject complex venom mixtures containing peptides ranging from < 10 to proteins > 260 kDa in size (Fig. S3). We detected traces of bacterial proteins in the venom proteome of

*E. nostras*, which are not considered reliable given the low number of peptides detected and the low intensity (Table S2). *C. carnea* venom contained few bacterial proteins including bacterial chaperonins (Table S3). The separation of *E. nostras* foregut and midgut extracts by gel electrophoresis showed distinct banding patterns compared to the venom, suggesting that the venom does not originate from the digestive tract (Fig. S3). In both species, ~60% of all venom protein candidates pre-selected by transcriptome analysis were also found in the venom proteomes (Fig. S4). The remaining 40% featured a signal peptide enabling secretion and were therefore also considered as putative venom proteins. Genes encoding proteins that were identified in the venom proteome were expressed in all head tissues, suggesting that the medial venom gland, the lateral venom gland and the cephalic gland are involved in venom production (Fig. 3). The overall relative protein composition appeared to be similar in both species, although we identified almost twice as many venom protein candidates in the *E. nostras* proteome (256 proteins) compared to the *C. carnea* proteome (137 proteins). Proteases were the most abundant functional class in both species, accounting for nearly 40% of all putative venom proteins (Fig. 3). In *E. nostras*, the maxillae showed a distinct expression profile, with multiple putative venom protein genes expressed strongly and specifically in this tissue (Fig. 3). In contrast, the expression profiles of the mandibles, anterior head and posterior head largely overlapped, with many non-specifically

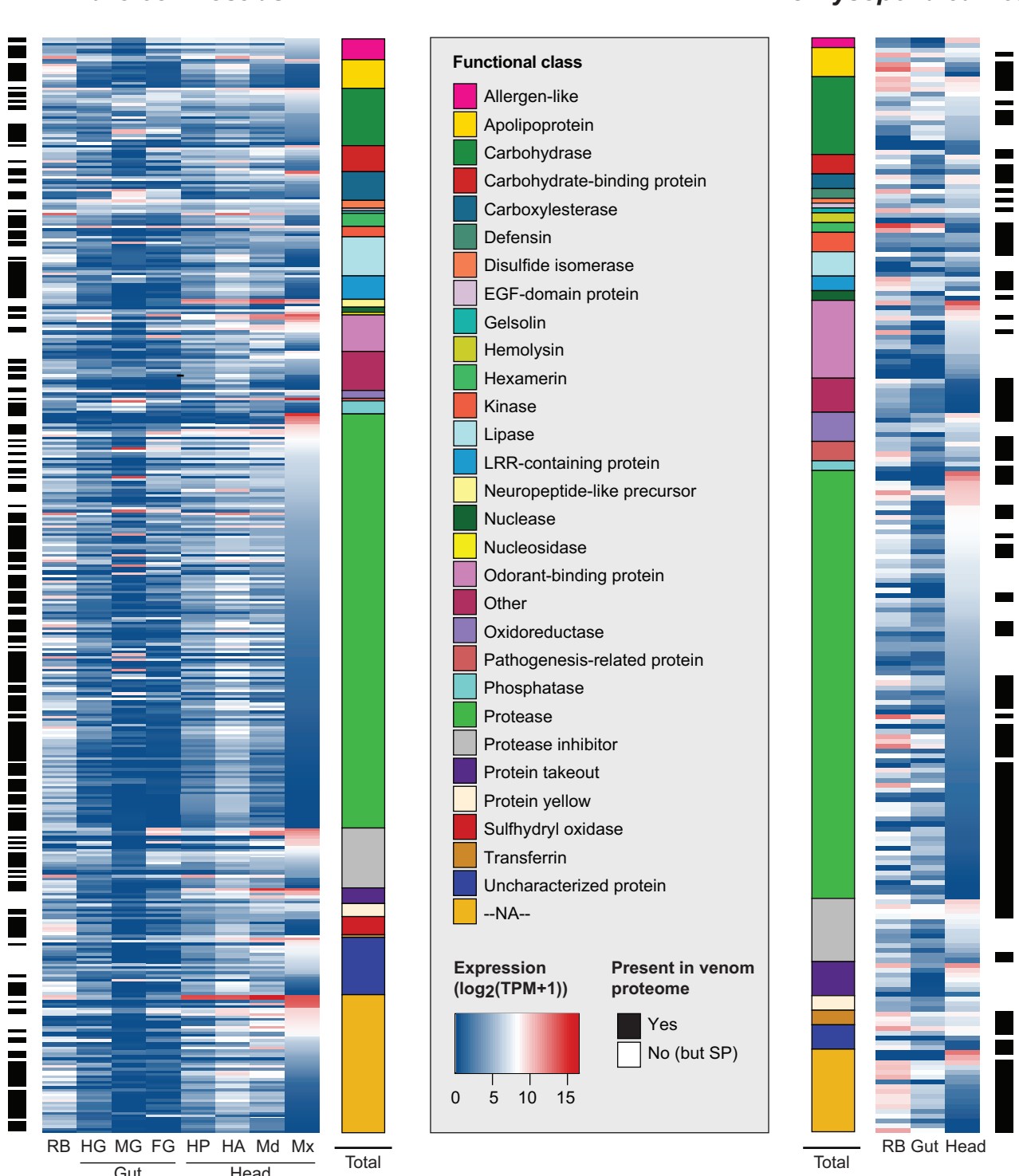

**Fig. 3 | Expression of putative venom proteins identified in the transcriptomes of *Euroleon nostras* and *Chrysoperla carnea*.** The heat maps represent the relative expression levels presented as $\log_2(\text{TPM} + 1)$ in the maxillae (Mx), mandibles (Md), anterior head (HA), posterior head (HP), foregut (FG), midgut (MG), hindgut (HG), and remaining body (RB) of *E. nostras*, and in the head, gut and RB of *C. carnea*. The identified contigs were grouped according to their protein family membership associations represented by color-coded blocks. The black bars next to the heat maps mark the proteins that were detected in the venom proteome. White bars represent proteins that were not found in the proteome but featured a signal peptide (SP).

expressed proteins (Figs. 3; S5). Most identified proteins could be assigned to protein families typically found in animal venoms such as S1 family peptidases, hemolysins and odorant-binding proteins. Some of the identified proteins, such as those containing leucine-rich repeats (LRRs),

represented families that are uncommon in animal venoms and their function in neuropterans is unclear. Despite the generally similar venom protein composition in *E. nostras* and *C. carnea*, we identified several proteins that were present solely in the *E. nostras* venom proteome,

including two phospholipases with strong expression in venom-related tissues (Figure S6). Moreover, we found 34 proteins in the *E. nostras* venom proteome that have no homology to known proteins in public databases. We aligned the coding sequences of these 34 proteins with the *E. nostras* and *C. carnea* genomes and found that all of them were encoded in the *E. nostras* genome, whereas 28 of them had no homologs in the *C. carnea* genome (Table S4).

### Differential expression of venom genes in the maxillary glands of E. nostras

To determine the glandular origins of neuropteran venom, we applied HCR-RNA-FISH to transversal sections of *E. nostras* using gene-specific probes that target mRNAs encoding putative venom peptides. From the genes that were most strongly expressed in the maxillae ($\log_2(\text{TPM} + 1) > 13$), we selected two genes broadly expressed across all tissues (*pr-5, na1*) and two genes specifically expressed in the maxillae (*vsp, na2*) (Fig. 4c). The full coding sequences of *pr-5, vsp* and *na1* were obtained from the *E. nostras* transcriptome, whereas the full coding sequence of *na2* was obtained from the *E. nostras* genome (Table S5). We found that the medial venom gland and lateral venom gland expressed different venom protein genes. The genes encoding a homolog of pathogenesis-related protein 5-like (PR-5) from *Leptinotarsa decemlineata* (Say, 1824) (Coleoptera, Chrysomelidae) and a protein lacking homology to known proteins (NA2) were strongly expressed only in the medial venom gland (Fig. 4a, d). In contrast, a gene encoding a homolog of Bi-VSP-like (VSP) from *Bombus impatiens* (Cresson, 1863) (Hymenoptera, Apidae) was strongly expressed in the lateral venom gland (Fig. 4a, d). Another gene encoding a protein lacking known homologs (NA1) was strongly expressed in epithelial cells, but not in the maxillary venom glands (Fig. 4a, d). Although NA1 was not detected in the venom by LC-MS/MS, we identified PR-5, NA2, and VSP in the venom proteome, confirming that parts of the secretions injected by *E. nostras* are produced in the medial venom gland and lateral venom gland. Moreover, HCR-RNA-FISH revealed that *pr-5, na2,* and *vsp* expression was weak or absent in non-venom tissues (Fig. S7), confirming the tissue-specific RNA-Seq expression profiles. In particular, *na2* was strongly and specifically expressed in the maxillae of *E. nostras* but was missing from the venom proteome, transcriptome, and genome of *C. carnea*. NA2 lacked any known structural motifs or protein domains other than the signal peptide, but the segment spanning amino acid residues 60–157 featured 14 glutamine-rich tandem repeats of six amino acids followed by a conserved proline residue (Fig. S8).

### Localization of potential bacterial symbionts

We used FISH with two general eubacterial probes to detect bacteria that may contribute to venom production and injection. We examined transverse sections of the head, thorax and abdomen, also focusing on organs associated with venom production and injection, such as the medial venom gland, lateral venom gland, cephalic gland, venom canal, and food canal. In *E. nostras*, we did not detect any intracellular or extracellular bacteria in any of these tissues (Fig. 5b, d). Furthermore, no bacteria were observed in tissues outside the venom system, such as the digestive tract, nervous system, exoskeleton, fat body, and hemolymph (Figure S9, S10). In *C. carnea*, numerous bacterial cells were found throughout the body, including extracellular bacteria in the gut lumen and fat body (Figure S11, S12) as well as intracellular bacteria in the medial and lateral venom gland (Fig. 5c, e, f) and the cephalic gland (Figure S11c).

### Insecticidal activity of neuropteran venoms

To compare the insecticidal activity of neuropteran venoms, we injected *Drosophila suzukii* (Matsumura, 1931) (Diptera, Drosophilidae) adult flies with head tissue homogenates of *E. nostras* and *C. carnea*, as well as *E. nostras* crude venom (Fig. 6). *Drosophila* species are part of the natural prey range of both *C. carnea* and *E. nostras*[37,38]. The *C. carnea* head tissue homogenate had no significant effect, whereas the *E. nostras* head tissue homogenate reduced the survival probability to 67% after 3 h and to 27%

after 24 h. The effect of 9.2 ng crude *E. nostras* venom was more potent, reducing the survival probability to 10% after 3 h and to 0% after 24 h. The *E. nostras* head tissue homogenate and crude venom also induced paralysis within 1 h whereas the *C. carnea* head tissue homogenate did not affect motility.

## Discussion

Neuropteran larvae are predatory insects that use venom to catch and pre-digest arthropod prey. They have adapted to a wide range of ecological niches, and antlions in particular are unique in their appearance and behavior. They catch relatively large prey using pit-fall traps and potent venom, but little is known about the tissue origin of the venom and its composition. We studied the morphology and biochemistry of the neuropteran venom system using larvae of the European antlion *E. nostras* and the green lacewing *C. carnea*. We demonstrated that both species produce a complex venom containing at least 137 (*C. carnea*) and 256 (*E. nostras*) proteins that are secreted from three different glands. To simplify the differentiation of these glands, we propose a new nomenclature, namely the medial venom gland, the lateral venom gland, and the cephalic gland (formerly the venom gland, lateral gland, and mandibular-maxillary gland, respectively). Only the venom from antlions conferred potent insecticidal activity against arthropod prey. Despite the general structural and compositional similarity of the venom systems, we found substantial inter-specific differences at the molecular level, which are likely to underpin the adaptation of *E. nostras* to its unique ecological niche.

The separation of the venom and food canal and the presence of specialized venom glands confirm that neuropteran larvae perform non-refluxing EOD, similar to true bugs and assassin flies[39,40]. We were able to collect clean predation venom from *E. nostras* and *C. carnea* using an artificial prey dummy. Proteotranscriptomic analysis showed that the venom proteins are produced by the medial venom gland and the lateral venom gland of the maxillae (Fig. 3; Fig. 4a, b), and a third secretory organ located in the head, which is probably the cephalic gland (Fig. 2a, e; Fig. 3). Whereas the medial and lateral venom glands are separated by the intrinsic muscle of the maxillary stylet and directly connect to the venom canal, the cephalic gland opens into the food canal (Fig. 2d, h). This indicates that part of the venom is injected through the venom canal and part through the food canal. We found that the medial venom gland and lateral venom gland secrete different proteins (Fig. 4a, b), indicating functional compartmentalization of the venom compounds. The spatial separation and context-dependent deployment of different venoms is not unique to arthropods. For example, the use of different venoms for predation and defense has been described in scorpions, spiders and true bugs[36,41−43]. Moreover, phytophagous true bugs use saliva from different gland compartments to form a salivary sheath and for EOD[44]. Other hypotheses that may explain functional venom compartmentalization include spatial confinement to prevent the destruction of toxins by digestive enzymes or the premature proteolytic activation of toxin precursors, and separation to allow the injection of different venom components at different time points. The ecological function of venom compartmentalization in neuropteran larvae is unclear, although the separation of predation and defensive venom seems unlikely because proteins from the medial venom gland, the lateral venom gland and the remaining head structures were detected in the predation venom. Moreover, it is improbable that the contents of the medial and lateral venom glands can be injected independently because they are both controlled by the same venom pump, which is formed by the intrinsic muscle of the maxillary stylet[45]. Although we cannot fully resolve the ecological role of venom compartmentalization in neuropteran larvae, our results shed light on the complexity of their venom system with three different glands secreting distinct protein mixtures.

The production of animal venom components by bacterial symbionts is rare. Centipedes have recruited several venom gene families from bacterial and fungal donors by horizontal gene transfer (HGT), but there is no evidence of direct symbiont involvement in toxin production[46]. Megalopygid caterpillars have acquired genes encoding aerolysin-like pore-forming

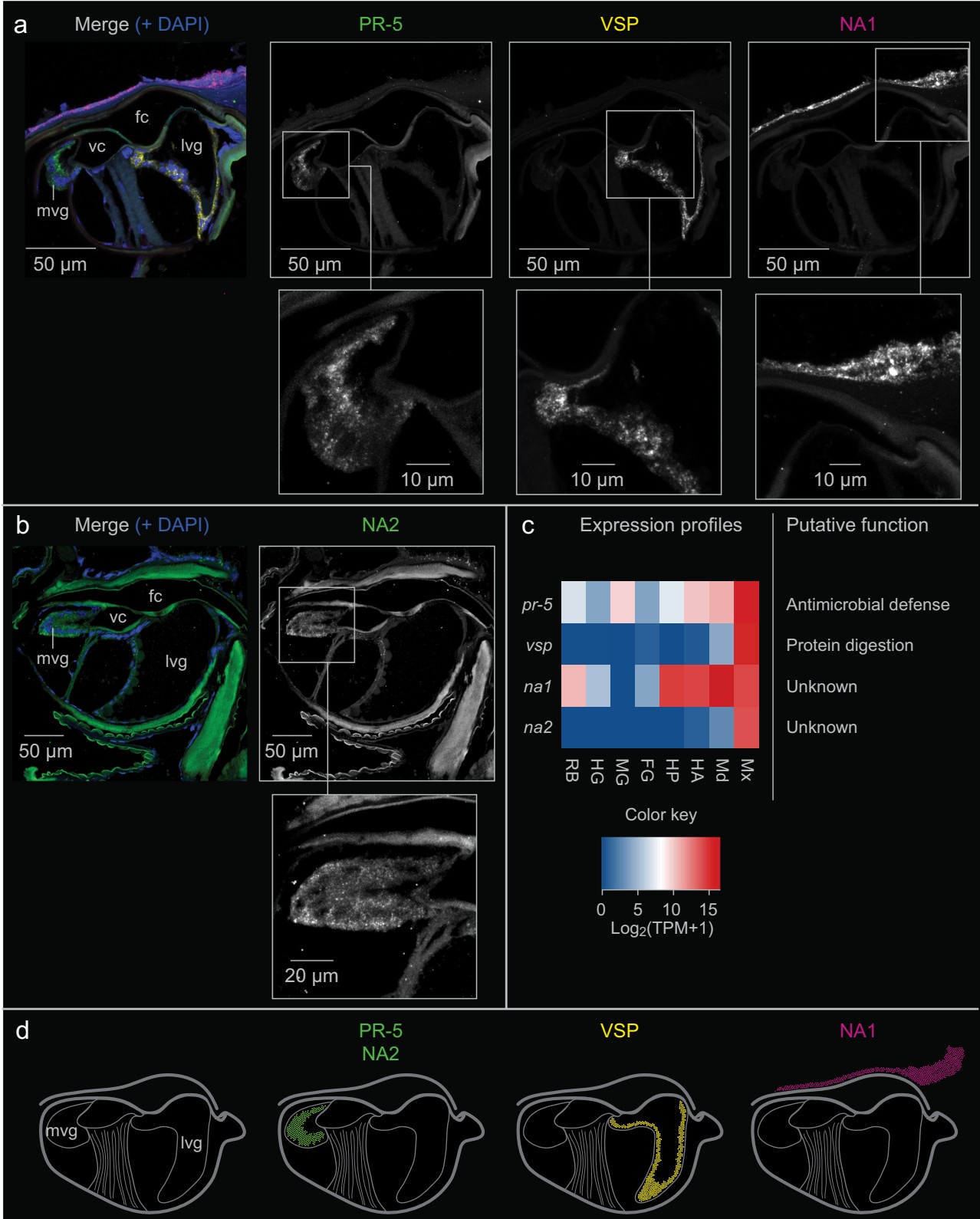

**Fig. 4 | Expression of selected genes encoding putative venom proteins in *Euroleon nostras*.** Cross sections were hybridized with gene-specific HCR-RNA-FISH probes for **a** *pr-5*, *vsp*, *na1* and **b** *na2*, alongside DAPI nuclear staining (blue). We found that *pr-5* (**a**) and *na2* (**b**) are expressed only in the medial venom gland (mvg), whereas *vsp* expression (**a**) is restricted to the lateral venom gland (lvg), and *na1* is not expressed in glandular cells but in epithelial cells instead (**a**). **c** RNA-Seq profiles of *pr-5*, *vsp*, *na1* and *na2* in the maxillae (Mx), mandibles (Md), anterior head (HA), posterior head (HP), foregut (FG), midgut (MG), hindgut (HG) and remaining body (RB), and their putative functions. **d** Schematic representation of the expression patterns detected for *pr-5*, *na2*, *vsp* and *na1* in the different tissues.

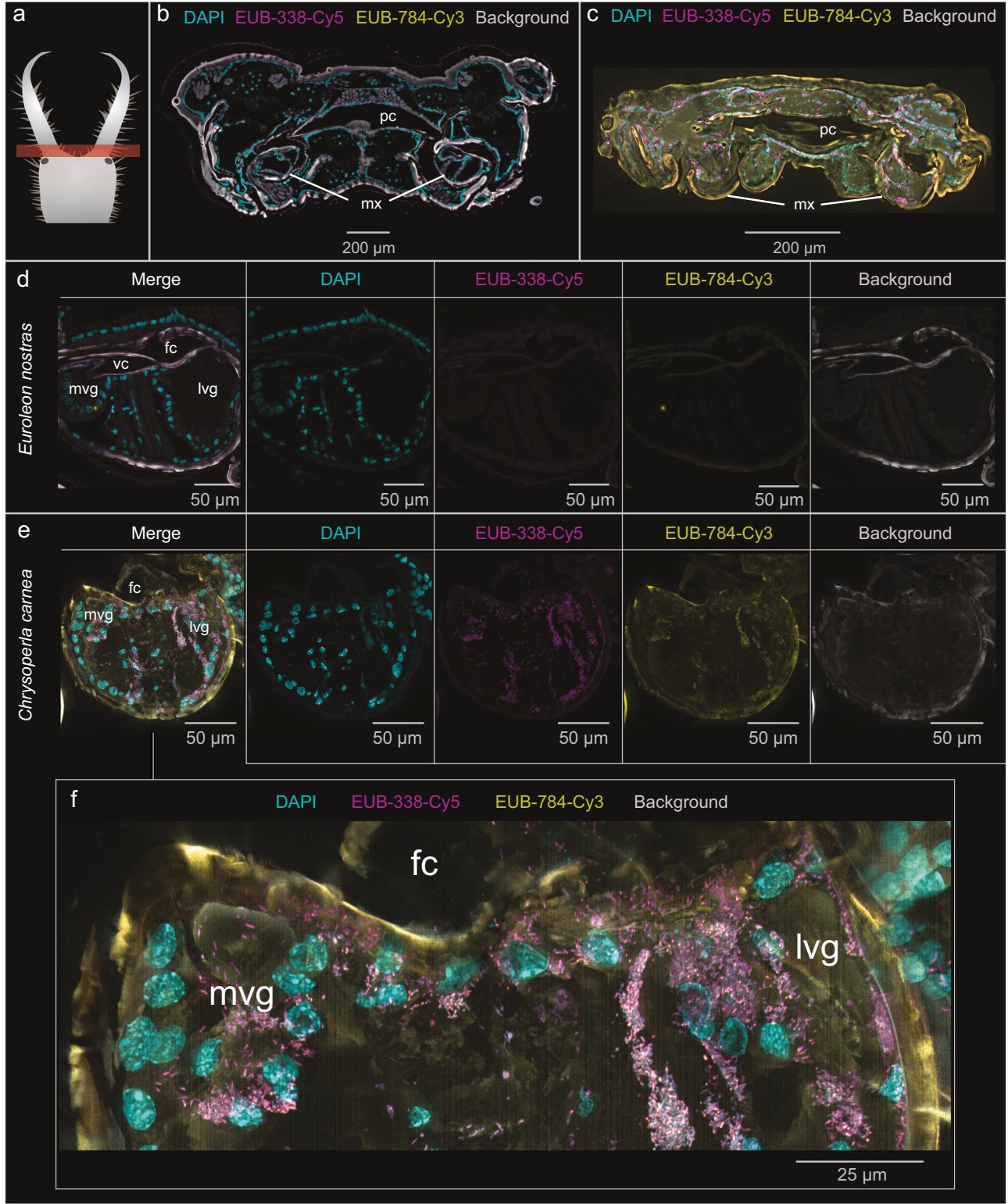

**Fig. 5 | Detection of bacteria in semi-thin transverse histological sections of *Euroleon nostras* and *Chrysoperla carnea* using FISH.** The labeled eubacterial probes EUB-338-Cy5 (magenta) and EUB-784-Cy3 (yellow) were used with DAPI to counterstain the nuclei (cyan). **a** The images show the area around the base of the mouthparts. **b** We did not detect any bacteria in *E. nostras* head tissues, including **d** the maxillae with the medial venom gland (mvg) and lateral venom gland (lvg). **c** In contrast, the head tissues of *C. carnea* contained numerous bacteria, including intracellular bacteria in the mvg and lvg (**e, f**).

toxins from bacteria also by HGT[47]. The first (and so far only) case of insect venom toxins produced by bacterial symbionts was reported in the antlion *M. bore*. In several studies, toxins were purified from the fermentation broth following the cultivation of bacteria isolated from antlion crops or esophageal tissue. The recovered proteins included a sphingomyelinase C from *Bacillus cereus*, a chaperonin (GroEL) from *Klebsiella aerogenes* (referenced as *Enterobacter aerogenes* in the original paper) and a sphaericolysin from *Lysinibacillus sphaericus*, which were shown to paralyze and

**Fig. 6 | Survival probability of *Drosophila suzukii* flies injected with PBS (negative control), 100% ethanol (positive control), *Euroleon nostras* and *Chrysoperla carnea* head tissue homogenates, or *E. nostras* crude venom.** Significant differences compared to untreated flies are indicated with asterisks (***$p < 0.001$; Cox proportional-hazards model, $n = 30$). Survival probability was reduced significantly by the injection of ethanol, *E. nostras* head tissue homogenate and *E. nostras* crude venom, but not *C. carnea* head tissue homogenate.

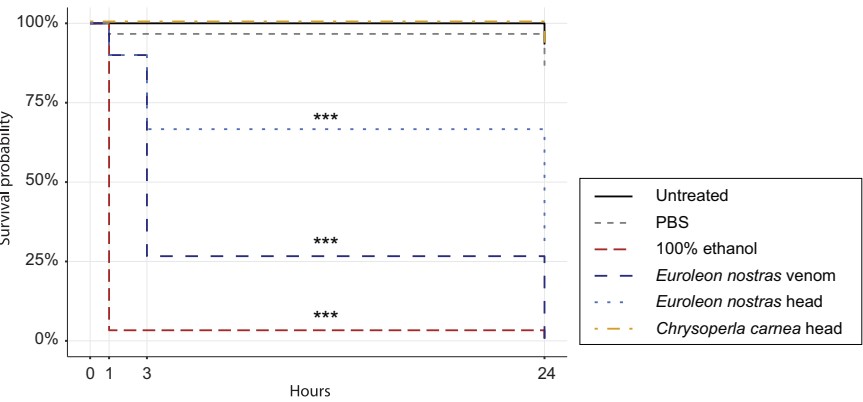

kill cockroaches[29,30,32]. The bacteria therefore appear to participate in a mutualistic relationship with the antlion, and produce venom toxins that help the host insect to overwhelm its prey[29,30,32]. Our results showed that neuropteran larvae do not inject secretions from the digestive tract into their prey, but have specialized venom-producing glands (Fig. 2b, f; Fig. 4). Moreover, we found that the venom system and digestive tract of *E. nostras* are devoid of extracellular and intracellular bacteria (Fig. 5b, d) and the venom contains no bacterial proteins. In contrast, *C. carnea* carried intracellular bacteria in the medial and lateral venom glands and the cephalic gland (Fig. 5f, Figure S11c). In consistence with this, *C. carnea* venom contained few bacterial proteins, including chaperonins such as GroEL (Table S3). Given that GroEL is one of the highest expressed proteins in obligate pathogens and symbionts and is particularly upregulated under stress conditions[48,49], it is not unexpected to find small amounts of it in the venom of *C. carnea*. It plays important roles in folding and unfolding of proteins but has also been suggested to be a major insecticidal toxin in the venom of the antlion *M. bore*[29,49]. However, we could show that *C. carnea* venom has no significant paralytic or lethal activity against insects, thus strongly indicating that neither GroEL nor other bacterial proteins in the venom of *C. carnea* act as insecticidal toxins. Our results therefore contradict assumptions that neuropteran venom toxins are produced by bacterial symbionts. The Firmicutes, which include *Bacillus* sp. and *Lysinibacillus* sp., and the Proteobacteria, including *Klebsiella* sp., are ubiquitous in the insect gut microbiome, where they fulfill multiple functions[50–53]. Even if they produce insecticidal compounds under aerobic conditions in vitro, this does not necessarily imply they do so under anaerobic conditions in the insect gut. There is no direct evidence based on in vivo experiments of a bacterial contribution to *M. bore* physiology and fitness. We cannot rule out species-dependent exceptions, but our results strongly suggest that neuropteran larvae generally inject insect-derived toxins produced in the venom glands rather than toxins produced by symbiotic bacteria in the gut.

The plasticity of venom composition plays a major role in ecological adaptation[54–57]. Neuropteran larvae have exploited a wide range of habitats and foraging strategies, probably accompanied by molecular adaptations[10,26,58]. Unlike the vegetation-rich environments inhabited by *C. carnea*, the harsh sandy habitats of *E. nostras* are characterized by high temperatures and scarce prey[9,21,23,24]. As a typical ambush predator, *E. nostras* has limited mobility and cannot afford to be selective in its prey choice, thus requiring potent, fast-acting venom to prevent prey escape and to extract maximum nutrients from even large prey. Accordingly, only *E. nostras* venom and head tissue homogenates caused the paralysis and death of *D. suzukii* flies within 24 h after injection, whereas *C. carnea* head tissue homogenates had no significant effect (Fig. 6). This suggests *E. nostras* venom plays a key role in prey capture whereas *C. carnea* is less reliant on potent venom as a predator of small, non-defensive prey. This is also reflected in the complexity of *E. nostras* venom, which contains twice as many proteins as the venom of *C. carnea*. *E. nostras* also secretes several

putative toxins to facilitate efficient prey capture that are not present in the venom proteome of *C. carnea*. We detected two strongly expressed phospholipase A2 (PLA$_2$) enzymes solely in the *E. nostras* venom proteome. PLA$_2$ is a common family of venom toxins that have been recruited independently into the venoms of multiple animals, including snakes, bees and true bugs[59]. They have diverse effects, including neurotoxicity, myotoxicity, hemolysis, anticoagulant, and insecticidal activities, and are often the main toxins in animal venoms[60,61]. The presence of PLA$_2$ in *E. nostras* but not *C. carnea* venom suggests that it plays an important role in the predation strategy of *E. nostras* and may contribute to the toxicity of its venom. Moreover, we detected multiple proteins in the highly potent *E. nostras* venom without homology to any known proteins and no homologous genes in the *C. carnea* genome (Table S4). This indicates the presence of orphan genes and thus the independent recruitment of novel genes into the venom protein repertoire of *E. nostras*. One of these undescribed venom effectors was NA2, one of the most abundant proteins in the maxillae of *E. nostras*. HCR-RNA-FISH showed that *na2* expression was restricted to the medial venom gland of the maxillae. The lack of homologous sequences in other arthropod species, including the *C. carnea* genome, suggests that NA2 is the product of an orphan gene recruited by *E. nostras* probably from non-coding DNA. In contrast to gene duplication and single gene co-option, the role of orphan genes in venom evolution is poorly understood and has been discussed mainly in robber flies and cone snails[62,63]. The species-dependent and tissue-specific expression of *na2* indicates venom-specific functions and may play a significant role in the adaptation of *E. nostras* to its unique ecological niche. Therefore, NA2 is an interesting candidate for the investigation of adaptive evolution in antlions and its function and activity should be studied in more detail. Our results, particularly the number of species-restricted venom-related genes in *E. nostras* suggest that the antlion's venom protein composition has evolved towards higher toxicity for prey insects than that of aphid lions, consistent with the differences in feeding ecology.

## Materials and methods
### Neuropteran larvae and venom collection
*Euroleon nostras* larvae were collected in Jena, Germany, and kept in containers filled with clean glass beads (diameter: 0.25–0.5 mm). They were fed L2–L4 stage *Spodoptera littoralis* (Boisduval, 1833) (Lepidoptera, Noctuidae) or *Chloridea virescens* (Fabricius, 1777) (Lepidoptera, Noctuidae) larvae once a week. To collect predation venom, the larvae were offered a prey dummy made of Parafilm and filled with 20 µl phosphate-buffered saline (PBS) as previously described[41]. The dummy was moved in the antlion pitfall traps until the larvae attacked. Larvae were allowed to inject venom for 10 min before the prey dummy was removed and the venom mixture was transferred to a clean microcentrifuge tube. *C. carnea* larvae were purchased from Katz Biotech and reared in separate containers on *Sitotroga cerealella* (Olivier) (Lepidoptera, Gelechiidae) eggs (Katz Biotech). Predation venom was collected as described for *E. nostras*. Venom collected from multiple *E. nostras* or *C. carnea* individuals was pooled and the protein concentration

was measured using an N60 nanophotometer (Implen). Venom samples were stored at –20 °C.

Foregut and midgut extracts were also collected from *E. nostras*. Larvae were anesthetized by chilling at –20 °C for 5 min before dissection in PBS. The foregut was separated from the midgut and placed in a separate tube containing 50 µL PBS and Halt Protease Inhibitor Cocktail (Thermo Fisher Scientific, 78429). Tissue samples from three individuals were pooled. The gut contents were extracted by vortexing for 5 s followed by low-speed centrifugation (1500 *g*, 3 min, 4 °C). The supernatant was further clarified by centrifugation (10,000 *g*, 5 min, 4 °C). The protein concentration was measured as above and gut extracts were stored at –20 °C.

### Insecticidal assays

Head homogenates of *E. nostras* and *C. carnea* were prepared by chilling the larvae at –20 °C for 5 min and decapitating them in PBS. The heads of five specimens of each species were pooled and homogenized in 50 µL PBS using a sterile pestle. Each homogenate was centrifuged (3000 g, 10 min, 4 °C) to remove cell debris and the supernatant was clarified by centrifugation (10,000 g, 10 min, 4 °C). *D. suzukii* flies from a laboratory stock (originally sourced from Ontario, Canada) were reared at 26 °C and 60% humidity with a 12-h photoperiod. They were fed a diet of 10.8% soybean and cornmeal mix, 0.8% agar, 8% malt, 2.2% molasses, 1% nipagin and 0.625% propionic acid. For the insecticidal assays, adult flies 4–6 days old were anaesthetized on a $CO_2$ pad (Inject+Matic) followed by the thoracic injection of 46 nL homogenate or crude venom using glass capillaries pulled on a P-2000 laser-based micropipette puller (Sutter Instrument) held on a Nanoject II device (Drummond Scientific). Injections were carried out under a Stemi 508 stereomicroscope (Zeiss). The protein concentration in the head tissue homogenates was 2.3 mg/mL and the venom concentration was 0.2 mg/mL. We injected 100% ethanol as a positive control and PBS as a negative control. We also maintained a cohort of untreated flies. Triplicate groups of 10 flies were injected and then reared in 29 × 95 mm vials with 30 × 30 mm foam stoppers (Nerbe Plus) filled 1/16 with food medium. Survival rates and paralysis were assessed for each specimen after 1, 3, and 24 h. Flies were considered dead if they showed no reaction or movement at all. In contrast, flies were considered paralyzed if they showed minimal movement but were unable to turn over on their own when flipped onto their backs. Statistical significance was determined using R v4.3.2[64] and RStudio v2023.6.2.561[65]. Survival probability was plotted using the *ggsurvfit* package[66] and analyzed using a Cox proportional-hazards model in the *coxme* package[67].

### Proteomic analysis

Venom and gut extracts were separated by sodium dodecylsulfate poly-acrylamide gel electrophoresis (SDS-PAGE) at 125 V on 4–12% Criterion XT gradient gels (BioRad, 3450123) using XT MES running buffer (BioRad, 1610789) for 1.5 h. Novex Sharp Pre-Stained Protein Standards (Thermo Fisher Scientific, LC5800) were run alongside the samples. After overnight staining with PageBlue protein staining solution (Thermo Fisher Scientific, 24620), the gels were washed in Millipore water until the background was clear. The gels were imaged using an Azure 600 Imaging System (Azure Biosystems).

The protein composition of *E. nostras* venom gel bands was determined by LC-MS/MS as previously described[41]. Briefly, proteins were digested in-gel with porcine trypsin[68] and the resulting peptides were separated by ultra-high performance liquid chromatography using an M-class system (Waters) coupled online to a Synapt G2-si mass spectrometer (Waters). Data-dependent acquisition (DDA) was carried out using MassLynx v4.1 software (Waters). MS-BLAST was used to search the Arthropoda database (downloaded from NCBI on 12 February 2019) and the *E. nostras* sub-database obtained by in silico translation of the transcriptome. In addition, the *pkl* files generated from raw data were used as queries against the NCBI nr database (downloaded on 12 February 2019) combined with the *E. nostras* sub-database using MASCOT v2.6.0.

We also analyzed the *E. nostras* and *C. carnea* venom proteomes in solution following the reduction of disulfide bonds with 10 mM

dithiothreitol in 50 mM HEPES (pH 8.5) for 30 min at 56 °C followed by the alkylation of free cysteines with 20 mM 2-chloroacetamide in 50 mM HEPES (pH 8.5) for 30 min at room temperature in the dark. After overnight digestion at 37 °C using sequencing-grade trypsin (Promega) in 50 mM ammonium bicarbonate (pH 8.5) at a ratio of 1:50, formic acid was added to a final concentration of 0.1% to stop the reaction. The samples were then desalted using an OASIS HLB µElution Plate (Waters) and dried by vacuum centrifugation. The samples were reconstituted in 10 µL of 1% formic acid and 4% acetonitrile and stored at –80 °C. An UltiMate 3000 RSLC nano LC system (Dionex) was fitted with a trapping cartridge (µ-Precolumn C18 PepMap 100, 5 µm, 300 µm i.d. x 5 mm, 100 Å) and an analytical column (nanoEase M/Z HSS T3 column 75 µm x 250 mm C18, 1.8 µm, 100 Å, Waters) and coupled to a QExactive Plus mass spectrometer (Thermo Fisher Scientific) using a Nanospray Flex ion source in positive ion mode. Trapping was achieved with 0.05% trifluoroacetic acid at a constant flow rate of 30 µL/min for 4 min on the trapping column. Peptides were eluted from the analytical column at a constant flow rate of 0.3 µL/min using increasing concentrations of solvent B (0.1% formic acid in acetonitrile): from 2% to 4% in 4 min, from 4% to 8% in 2 min, from 8% to 25% in 41 min, from 25% to 40% in 5 min, and from 40% to 80% in 4 min. The peptides were introduced into the mass spectrometer via a Pico-Tip Emitter (MSWIL) with an applied spray voltage of 2.2 kV, and a capillary temperature of 275 °C. Full mass scans were acquired in profile mode within a mass range of 350–1500 *m/z*, a resolution of 70,000, and a maximum filling time of 100 ms with a limitation of $3 \times 10^6$ ions. For DDA, the Orbitrap resolution was set to 17,500 with a filling time of 50 ms and a limitation of $1 \times 10^5$ ions. A normalized collision energy of 26 eV was applied, with a loop count of 20, an isolation window of 1.7 *m/z*, and a dynamic exclusion time of 20 s. The peptide match algorithm was set to 'preferred' and charge exclusion to 'unassigned', excluding charge states 1, 5–8, and > 8. MS/MS data were acquired in centroid mode. The raw data were processed using Frag-Pipe v21.1 with MSFragger v4.0[69] and searched against sub-databases based on the in silico translation of the *E. nostras* and *C. carnea* transcriptomes and *C. carnea* genome, merged with the SwissProt database (downloaded on 18 January 2023). Common contaminants were included in the search and decoy mode was set to revert. Carbamidomethylation of cysteine was considered as a fixed modification, whereas oxidation of methionine and N-terminal acetylation were considered as variable modifications. The mass error tolerance was set to 20 ppm for full scan MS spectra and 0.5 Da for MS/MS spectra. A maximum of two missed cleavages was allowed. For protein identification, a minimum of one unique peptide with a minimum length of seven amino acids and a false discovery rate < 0.01 were required at both the peptide and protein levels.

### Transcriptomic analysis

Larval tissues from *E. nostras* (mandibles, maxillae, anterior head, posterior head, crop, midgut, hindgut, rest of body) and *C. carnea* (head, gut, rest of body) were dissected in PBS and immediately transferred to ice-cold Trizol (Zymo Research, R2050-1-200). Tissues from five (*E. nostras*) and six (*C. carnea*) specimens were pooled and homogenized in ceramic bead tubes using a TissueLyser LT (Qiagen). Total RNA was extracted using the Direct-zol RNA Miniprep kit (Zymo Research, R2052). The concentration and integrity of the RNA were determined using an N60 nanophotometer and a TapeStation System (Agilent), respectively. PolyA+ mRNA was enriched from 1 µg total RNA using oligo-dT magnetic beads and fragmented to an average of 250 bp. The TruSeq RNA library preparation kit was used to generate sequencing libraries. Sequencing of one replicate pool per species and tissue was carried out by the Max-Planck Genome Center, Cologne, on an Illumina HiSeq3000 Genome Analyzer platform using paired-end (2 × 150 bp) read technology. The reads were processed using an in-house assembly and annotation pipeline including the filtering of high-quality reads, removal of reads containing primer/adaptor sequences, and trimming of read lengths using CLC Genomics Workbench v11.1. For transcriptome assemblies, all tissue samples from one species were combined. The de novo assemblies were created using CLC Genomics Workbench

v11.1 standard settings and two additional CLC-based assemblies with different parameters. The presumed optimal consensus transcriptome was selected as previously described[70]. Annotations were added using BLAST, Gene Ontology and InterProScan in OmicsBox (https://www.biobam.com/omicsbox) as previously described Götz, et al.[71]. Up to 20 of the best non-redundant hits per transcript were retained for BLASTx searches against the NCBI nr protein database using an e-value cutoff of $\leq 10^{-3}$ and a minimum match length of 15 amino acids. Transcriptome completeness was assessed by benchmarking universal single-copy orthologs (BUSCO) analysis by comparing the assembled transcriptome against a set of highly-conserved single-copy orthologs. We used the BUSCO v3 pipeline[72] to compare the proteins predicted from the *E. nostras* and *C. carnea* transcriptomes with the predefined set of 1658 Insecta single-copy orthologs from the OrthoDB v9.1 database (Table S6). Digital gene expression analysis was implemented in CLC Genomics Workbench v11.1 by generating BAM files and using sequence counts to estimate expression levels, with previously described parameters for read mapping and normalization[73]. Mapped read values were normalized as implemented in CLC Genomics Workbench v11.1 and ArrayStar v17, and used to calculate log$_2$-transformed transcripts per million (TPM + 1) values. Venom protein candidates were selected in the transcriptome according to venom-specific annotations, tissue-specific expression and the presence of a signal peptide for secretion. The preselected putative venom proteins were complemented with the venom proteome, and only candidates that were present in the proteome and/or had a signal peptide were retained.

## Genomic analysis

The *E. nostras* genome sequence was assembled from PacBio HiFi data generated from a single adult female specimen. High-molecular-weight genomic DNA was extracted using the Nanobind Tissue Big DNA kit (Circulomics, NB-900-001-01), and SMRTbell HiFi libraries were prepared and sequenced on a Pacbio Sequel IIe instrument (Pacific Biosciences) at the Max Planck Genome Center in Cologne, Germany. Hifiasm v0.16.0 was used for genome assembly, followed by an additional PurgeDups step to remove duplicates and produce a haploid genome. The final genome assembly comprised 52 contigs with a total size of 1.48 Gb and a contig N50 length of 97.6 Mb. The BUSCO assessment indicated a completeness of 98.6% of single-copy Insecta orthologs (Table S6). For *C. carnea*, a publicly available genome assembly (GenBank accession GCA_905475395.1) was downloaded from the NCBI. Selected venom proteins from *E. nostras* were searched against the *E. nostras* and *C. carnea* genomes using BLAST v2.15.0 to identify antlion-specific venom toxins. The e-value cut-off was set to 0.01. Genes also found in the *C. carnea* genome were then also searched against the *C. carnea* transcriptome and checked for their presence in the *C. carnea* venom proteome.

## Histology

For structural analysis, *C. carnea* and *E. nostras* larvae were immersed in water to remove sand and impurities, then transferred to ice-cold 0.2 M phosphate buffer (pH 7.2) and fixed with 2.5% glutaraldehyde in phosphate buffer for 1 h. The samples were post-fixed in 1% osmium tetroxide in phosphate buffer at room temperature for 1 h. After dehydration through a graded ethanol series, the larvae were embedded in Araldite epoxy resin (Plano, R1040) and semi-thin sections were prepared using a Reichert Om/U3 ultra-microtome (Leica Microsystems). The sections were stained with 0.5% toluidine blue in 0.5% sodium tetraborate (prepared in distilled water) and viewed using a DM 4 B microscope (Leica Microsystems).

## µCT

We used µCT analysis to reconstruct the *E. nostras* and *C. carnea* larval venom systems. Specimens were fixed in 4% paraformaldehyde (PFA) in 80% ethanol for 48 h. The samples were washed twice in 80% ethanol for 1 h each and in denatured ≥ 99.8% ethanol for 24 h. After incubation in 1% iodine in methanol for 24 h, the samples were washed three times in denatured ≥ 99.8% ethanol and three times in pure ethanol for 1 h each. The

samples were then dried in an EM CPD300 critical point dryer (Leica Microsystems) with a slow-speed CO$_2$ supply, a delay of 60 min, 35 exchange cycles, heating at medium speed, and slow gas exhaust. The dried samples were mounted on specimen holders using UV adhesive and fishing line. We applied X-ray scans in a SkyScan 1272 µCT (Bruker) using 360° rotation with 0.2-µm rotation steps. The voltage and current were adjusted to reach a minimum transmission of 30–50%. Alignment, ring artifact correction, and 3D reconstruction were controlled using NRecon v2.0.0.5 software (Bruker). The reconstructions were analyzed using Dragonfly 2022.2 for Windows (Comet Technologies Canada Inc.; software available at https://www.theobjects.com/dragonfly).

## FISH

Three larvae per species were fixed in 80% *tert*-butanol containing 4% PFA for 48 h, washed four times by agitating in 80% *tert*-butanol for 10 min each and dehydrated using increasing concentrations of *tert*-butanol (90%, 96%, 3×100%) followed by acetone (3 × 100%) for 2 h each[74]. After infiltration and embedding in HistoCure 8100 (Morphisto, 12226), transversal semi-thin sections (8 µm) were obtained using an RM2245 rotation microtome (Leica Microsystems) with glass knives. The sections were mounted on microscope slides and hybridized with 500 nM of the general eubacterial probes EUB-338-Cy5 (5′-GCT GCC TCC CGT AGG AGT-3′) and EUB-784-Cy3 (5′-TGG ACT ACC AGG GTA TCT AAT CC-3′) in 100 µL hybridization buffer (0.9 M NaCl, 0.02 M Tris-HCl pH 8.0, 0.01% SDS) containing 0.05 mg/mL DAPI in a humid chamber at 50 °C overnight. The sections were washed (0.9 M NaCl, 0.02 M Tris-HCl pH 8.0, 0.01% SDS, 5 mM EDTA) for 60 min at 50 °C and rinsed in distilled water for 20 min at room temperature. The sections were mounted under high-precision coverslips (Paul Marienfeld) in VectaShield (Vector Laboratories, H-1000-10) and imaged on an inverted Dmi8 Thunder Imaging System (Leica Microsystems) using the Leica Application Suite X software.

## HCR-RNA-FISH

Two *E. nostras* larvae were fixed as described above for standard FISH and then pre-embedded in 1% agar before washing and dehydration as described above but with the 100% acetone steps replaced with 100% isopropanol. After infiltration with two batches of paraffin at 60 °C (2 and 12 h, respectively), the larvae were embedded in paraffin[74], and transversal semi-thin sections (5 µm) were prepared on the Leica RM2245 rotation microtome with disposable blades. The sections were deparaffinated, post-fixed in 4% PFA for 20 min, then digested with pepsin (0.4% in 0.9% NaCl, pH 1.5) for 15 min at 37 °C. Gene-specific HCR v3.0 probes (20 pairs per gene, or as many as possible for short mRNA sequences) and labeled hairpins were obtained from Molecular Instruments. Buffers were prepared as recommended by Molecular Instruments with one exception: the amount of dextran sulfate in the probe hybridization/amplification buffers was changed from 10% to 5% (v/v) to reduce viscosity. The following combinations of amplifier and fluorophore were used: B1 and Alexa-Fluor (AF) 488 for *pr-5*, B2 and AF 546 for *vsp*, and B3 and AF 647 for *na1*. The expression of *na2* was examined in a separate experiment with the B1 amplifier and AF 488. HCR-RNA-FISH was performed according to the Molecular Instruments HCR protocol for generic samples on slides (https://files.molecularinstruments.com/MI-Protocol-RNAFISH-GenericSlide-Rev9.pdf), except that 400 µL of probe hybridization buffer was used for pre-hybridization instead of 200 µL and the amount of probe per slide was doubled to 0.8 pmol. After amplification but before washing, nuclei were counterstained with 100 µL 1 µg/mL DAPI in 5x SSCT for 1 h in a dark, humidified chamber at room temperature. The sections were mounted under high-precision coverslips in ProLong Glass mounting medium (Thermo Fisher Scientific, P36980). Confocal images were captured on an LSM 880 confocal microscope (Zeiss) and 16-bit image stacks were taken in a 1024 × 1024 scan format with 4× averaging, a speed setting of 6, and a z-step size of 1 µm. A C-Apochromat 10×/1.2 M27 water immersion objective was used for overview images, whereas Plan-Apochromat 20×/0.8 M27 air immersion and C-Apochromat 40×/1.2 M27 water immersion

objectives were used for close-ups. Imaging channels were acquired semi-sequentially (DAPI and AF-546 or AF-594 were imaged together, as were AF-488 and AF-647) to reduce crosstalk. The following settings were used: DAPI, 405 nm excitation, 415–510 nm emission; AF-488, 488 nm excitation, 500–550 nm emission; AF-546, 543 nm excitation, 560–590 nm emission; AF-594, 543 nm excitation, 600–630 nm emission; and AF-647, 633 nm excitation, 650–690 nm emission.

## Reporting summary

Further information on research design is available in the Nature Portfolio Reporting Summary linked to this article.

## Data availability

The proteomic data have been deposited in the ProteomeXchange Consortium (http://proteomecentral.proteomexchange.org) via the PRIDE partner repository[75] with the dataset identifier PXD047026. The transcriptomic data have been deposited in the European Nucleotide Archive (ENA) at EMBL-EBI under accession number PRJEB70475. The genomic dataset for *E. nostras*, the transcriptome assemblies and the sequences of the selected venom protein candidates for both species have been deposited in the Edmond data repository and can be accessed under the following weblink: https://doi.org/10.17617/3.YZVPXB[76]. Detailed information on sequences, annotations, and gene expression for venom protein candidates of *E. nostras* and *C. carnea* can be found in Supplementary Data 1 and 2, respectively. The data underlying the insecticidal assay shown in Fig. 6 are provided in Supplementary Data 3. All other data are available from the corresponding author on reasonable request.

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

## Acknowledgements

The authors thank Richard M Twyman for manuscript editing and the Max Planck Society for funding. AV acknowledges generous funding from the Hessian Ministry of Higher Education, Research and the Arts for the project Animal Venomics embedded in the LOEWE Centre for Translational Biodiversity Genomics (LOEWE-TBG).

## Author contributions

Conceptualization: H.V., A.V., M.L.F.; Investigation: M.L.F., H.S., O.T., B.W., L.D., T.L., N.W.; Formal analysis: M.L.F., H.V.; Visualization: M.L.F., H.S.; Funding acquisition: H.V., A.V.; Supervision: H.V., A.V., M.K.; Writing–original draft: M.L.F., H.S., O.T., B.W.; Writing—review and editing: M.L.F., H.S., O.T., B.W., T.L., N.W., A.V., M.K., H.V.

## Funding

## Competing interests

The authors declare no competing interests.
