## [Peer Review File · Communications Biology]

Reviewers' comments:

Reviewer #1 (Remarks to the Author):

The authors have undertaken a comparative study of the venom system and venom of two ecologically divergent neuropteran species. This work includes a study of the morphology of the venom system, venom composition, insecticidal activity, glands involved in toxin production, and the possible involvement of symbiotic bacteria in toxin synthesis.

If the question is whether this manuscript advances the knowledge of neuropteran venoms and merits publication, the answer is definitely yes. The amount of data collected by the authors is impressive and the work is ambitious. To my knowledge, this is the most important study of the venom of these insects.

However, I have several major concerns with the manuscript that should be addressed.

I- There are many very general statements throughout the article that suggest evolution and that this work reveals adaptive mechanisms to different ecological niches. All of these statements are unsupported or insufficiently supported by the results presented, and need to be strongly moderated (including the title) for the following reasons:

- First, the authors rely solely on a comparison between two species to draw far too general conclusions.

- Second, the authors rely on the results of insecticidal activity and venom compositions for different to support their point. The insecticidal part is not convincing. The authors tested only one crude venom on both species. For *C. carnea*, the authors base their claim that the venom is not insecticidal on a head extract while the toxins may be too diluted in this extract. In addition, the insecticide assays were conducted on a model insect (*Drosophila suzukii*) that seems far from the natural prey. What's more, the authors speak of insecticidal activity, whereas *C. carnea* venom could be paralyzing but not lethal. In the M&M, the authors refer to paralysis data that have been obtained, but these data are absent from the manuscript, although they could provide real added value to the results and, above all, greater ecological relevance (the aim of an offensive venom is first and foremost to paralyze prey not to kill). Nor do we know how the M&M authors distinguished between dead, paralyzed and surviving insects.

- Finally, it seems rather difficult to get an idea of the venom composition of the two species. Based on Figure 3, the compositions don't seem all that different. It might have been interesting to show only the proteins confirmed by proteomics in a pie chart. In the discussion, the authors discuss proteins that are unique to *E. nostras*, but the biological function of most of these proteins is unknown. Regarding PLA2, the authors attribute an offensive function to this toxin to paralyze prey, whereas in many studies PLA2 has a predominantly defensive function (e.g., it is present in exclusively defensive venoms such as that of honey bees). All these data do not allow us to conclude that the "dynamics of venom protein composition play an important role in the ecological diversification of neuropteran larvae".

II- In the M&M we learn that the authors have produced the genome of *E. nostras*. This is remarkable, but the genome seems to have very little use in the manuscript. It is not described at all in the results, nor does it seem to have been used to map the transcriptomic reads. Furthermore, the genomic data have not been deposited in a public database.

Reviewer #2 (Remarks to the Author):

Review of COMMSBIO-24-2102-T

General comments

This manuscript describes a large body of work examining the production, composition, and activity of venoms produced by the Neuroptera (lacewings, antlions, and allies). As far as I am aware it is the first study to do so and as such it is a very important contribution to the field of arthropod venoms. There is a substantial amount of confusion in the literature to do with neuropteran venoms—some to do with the anatomy of venom production and some to do with supposed toxins produced by hypothesised bacterial symbionts. The objective approach of the current study clears away a lot of this confusion and that is very welcome.

I do however think the manuscript could be substantially improved. The most important things are to clarify if it is known if the glands in the head (the maxillary-mandibular glands) could (considering the anatomy) or do contribute to the venom or if this is just unknown—I couldn't quite work this out from reading the current manuscript. As far as I can tell, the glands called the 'venom gland' and 'lateral gland' definitely do contribute to the venom, and if this is right I'd suggest they are called 'medial venom gland' and 'lateral venom gland' throughout. The 'maxillary-mandibular gland' name is currently quite confusing and it could be called something else e.g. 'cephalic gland'. I'd encourage the authors to discuss how each of these glands corresponds to glands or nomenclature identified/used by previous authors (include citations) but also introduce a sensible, useful nomenclature that will simplify future studies of neuropteran venoms.

One section of the manuscript examines the presence of bacteria, presumably because of the previous studies of supposed bacterial endosymbionts. The methods used here are kind of cool but the results are actually not of high relevance on reflection and I suggest the authors consider if it is really necessary or if it could be removed or transferred to supplementary material. Confusingly, one species had plenty of bacteria and one had none, but this prompts us to ask what the relevance of this outcome is. If we believe the result from the species that has no bacteria associated (and I'm not sure I do because it's hard to see what the positive control would be and I've never heard of an insect with no microbiome), it suggests the endosymbionts are not important for venom function. For the species with numerous associated bacteria, does the result suggest that bacterial endosymbionts are important for venom function? No. I think the proteomic results and the annotation of the proteins present in venom as observed by MS being of insect origin are the important results from which it can be concluded that the previous studies on toxins produced by bacterial endosymbionts are not relevant to neuropteran biology. This comes with the caveat that to really make this result airtight, the MS data should be searched against databases including bacterial sequences and I don't think that was done.

Overall I recommend this important manuscript is accepted after revisions in response to the

issues I have raised.

Specific issues

- Line 29: “The Neuroptera is” should be either “Neuroptera is” or “The Neuroptera are”
- Line 57: “There is a lack of molecular data to confirm such symbiotic relationships”. In the light of this study, this statement is kinder and more vague than is warranted. Consider adding a sentence to the effect of “However, these previous studies did not demonstrate the occurrence of these bacterial toxins in venom or demonstrate they have any relevance to natural prey capture by neuropterans”.
- Line 97: What is the rationale for calling one a venom gland and the other a lateral gland? If this nomenclature is based on previous studies please cite them.
- Line 119: “peptides ranging from < 10 to > 260 kDa in size” should be “peptides ranging from < 10 to proteins > 260 kDa in size”
- Line 122: Please elaborate in more detail the ‘transcriptome analysis’ used to find these other 40% of putative toxins
- Line 133 comment: There are several proteins shared with true bugs, who have similar requirements for extraoral digestion, including e.g. gelsolin which you can imagine contributing to EOD
- Line 139: Could you reword to clarify you mean these proteins do not have homologues in the *C. carnea* genome?
- Line 162: Please indicate how or why you selected these genes/proteins for analysis.
- Line 177: Again, if the lateral gland and venom gland are both contributing to the venom should you reconsider the nomenclature? Should they be lateral venom gland and medial venom gland?
- Line 178: I can’t find NA2 in the spreadsheet, not sure why? The manuscript says it was detected in venom but searching either the name or parts of the sequence shown in Figure S7 yield nothing.
- Line 198: It’s weird that no bacteria at all were found in any tissues (including the exterior of the animal and the gut). Did these experiments just not work? A control here would have been helpful to interpret this experiment. But also, I don’t really find the results of this experiment that relevant. If there are bacteria there or not, what matters is what is in the venom, which you have shown is encoded by insect genetic material. I almost feel like this paper would be benefit from removal of this section on localisation of microbes and addition of a statement stating this finding explicitly (e.g. at the end of the section titled Protein composition of neuropteran venoms, you could state “These results demonstrate that neuropteran venom toxins are encoded by insect genetic material and not bacterial symbionts as previously hypothesised (cite papers).”
- Line 221: 3 h seems like a pretty long timepoint for a venom that is meant to be used for prey capture to first show effect, could the authors comment on this?
- Line 244: I thought the stuff in the venom was from two different glands, the ‘venom gland’ and ‘lateral gland’ but not the maxillary-mandibular gland which opens into the food canal. But it says three here?
- Line 251: Should probably mention EOD in the Introduction somewhere
- Line 255: This suggests the maxillary-mandibular gland contributes to venom but I’m unsure what the evidence for this is. I think the way the Results is written and presented should be altered to make a few things more clear: (a) Do you expect the maxillary-mandibular gland to be in the head

RNA-Seq sample but the venom gland and lateral gland in the maxillae RNA-Seq sample? (b) Can you do a statistical analysis to ask the question of if venom proteins (those detected by LC-MS/MS) are more highly expressed in the maxillae RNA-Seq sample than the head RNA-Seq sample. If yes, explicitly argue that this result suggests venom is made in the venom gland and lateral gland but not the maxillary-mandibular gland. Or if not, say that it means the maxillary-mandibular gland likely contributes to venom production. (c) In light of the anatomy elucidated in this paper, does it make sense that the maxillary-mandibular gland contributes to venom production?

- Line 325: In what sense are these genes orphans?
- Line 331: Why probably from non-coding RNA?
- Line 399: Wouldn't you usually use data independent acquisition in this type of study?
- Line 402: Would these methods have detected toxins produced by bacterial endosymbionts if present? If not, should the MS files be searched against bacterial databases to check?
- Line 410: Should you give a pH value for the amm bicarb?
- Line 463: Are the results from these BUSCO comparisons in the manuscript or supp data somewhere?
- Line 473: I think it would be better to only call those peptides and proteins that were detected in venom putative venom toxins (throughout the manuscript) but if the authors think differently that is fine.
- Line 475: I didn't realise this whole genome was assembled during the current study until now, maybe make this clearer with a sentence or two in the Results or elsewhere
- Line 566: The link isn't live yet, but does the transcriptomic data deposited include both the raw reads and assembled transcriptome?
- Has the genomic data also been deposited somewhere?

Reviewers' comments:

Reviewer #1 (Remarks to the Author):

The authors have undertaken a comparative study of the venom system and venom of two ecologically divergent neuropteran species. This work includes a study of the morphology of the venom system, venom composition, insecticidal activity, glands involved in toxin production, and the possible involvement of symbiotic bacteria in toxin synthesis.

If the question is whether this manuscript advances the knowledge of neuropteran venoms and merits publication, the answer is definitely yes. The amount of data collected by the authors is impressive and the work is ambitious. To my knowledge, this is the most important study of the venom of these insects. However, I have several major concerns with the manuscript that should be addressed. I- There are many very general statements throughout the article that suggest evolution and that this work reveals adaptive mechanisms to different ecological niches. All of these statements are unsupported or insufficiently supported by the results presented, and need to be strongly moderated (including the title) for the following reasons:

- First, the authors rely solely on a comparison between two species to draw far too general conclusions.

We acknowledge the important points the reviewer raised and have made the following changes:

We changed the title to “Divergent venom effectors correlate with ecological niche in neuropteran predators”.

Lines 23-25: “Our results indicate that molecular venom evolution plays a role in the adaptation of antlions to their unique ecological niche.”

Line 362-365: “Our results, particularly the number of species-restricted venom-related genes in *E. nostras* suggest that the antlion’s venom protein composition has evolved towards higher toxicity for prey insects than that of aphid lions, consistent with the differences in feeding ecology.”

- Second, the authors rely on the results of insecticidal activity and venom compositions for different to support their point. The insecticidal part is not convincing. The authors tested only one crude venom on both species. For *C. carnea*, the authors base their claim that the venom is not insecticidal on a head extract while the toxins may be too diluted in this extract. In addition, the insecticide assays were conducted on a model insect (*Drosophila suzukii*) that seems far from the natural prey. What's more, the authors speak of insecticidal activity, whereas *C. carnea* venom could be paralyzing but not lethal. In the M&M, the authors refer to paralysis data that have been obtained, but these data are absent from the manuscript, although they could provide real added value to the results and, above all, greater ecological relevance (the aim of an offensive venom is first and foremost to paralyze prey not to kill). Nor do we know how the M&M authors distinguished between dead, paralyzed and surviving insects.

We thank the reviewer for the comments and suggestions, but we politely disagree with the reviewer’s characterization of the insecticidal assays. We mainly used head homogenates because it is very difficult to obtain significant quantities of venom, particularly from *C. carnea*. For *E. nostras*, we also used venom collected using a prey dummy to support our results, as venom collection is feasible in this species. We assessed survival rates and paralysis for both species. Dead flies showed no movement or reaction at all,

whereas paralyzed flies showed minimal movement but were unable to turn over on their own when placed on their backs. Unaffected flies were mobile, showed typical feeding and grooming behavior, and turned over immediately when placed on their backs. We have added this information to the materials and methods section for clarity:

Lines 408-411: “Flies were considered dead if they showed no reaction or movement at all. In contrast, flies were considered paralyzed if they showed minimal movement but were unable to turn over on their own when flipped onto their backs.”

The *E. nostras* venom and head homogenates were both lethal and paralytic, whereas *C. carnea* head homogenates conferred neither effect. We refer to the paralytic activity in the results (**Lines 244-246**). The head homogenates of *E. nostras* and *C. carnea* are comparable, as the same amounts (46 nL of a 2.3 mg/mL concentrated homogenate) were injected. Given the ecological differences between *E. nostras* and *C. carnea* – with *E. nostras* being able to overwhelm large, well-defended insects and *C. carnea* feeding mostly on small, soft-bodied insects – the lack of paralytic and lethal activity is not surprising. This is also consistent with our observations while feeding *E. nostras* and *C. carnea*: we found that *E. nostras* usually paralyzes even large prey within seconds to minutes, whereas *C. carnea* only pierces its prey with strong pincers and sucks out the contents without signs of paralysis. We are therefore confident that the results of the insecticidal assays are accurate and relevant.

Although *C. carnea* feeds mostly on aphids, it can also attack and feed on a variety of other insects, including *Drosophila suzukii* (Englert C, Herz A (2019) Acceptability of *Drosophila suzukii* as prey for common predators occurring in cherries and berries. J Appl Entomol 143, 387-396). We therefore do not agree that *D. suzukii* makes the insecticidal activity assay results less relevant. However, we have added the following information in the results section to justify the use of this model prey:

Lines 233-234: “*Drosophila* species are part of the natural prey range of both *C. carnea* and *E. nostras*^{37, 38}.”

37. Bonneau P, Renkema J, Fournier V, Firlej A. Ability of *Muscidifurax raptorellus* and other parasitoids and predators to control *Drosophila suzukii* populations in raspberries in the laboratory. *Insects* 10, 68 (2019).
38. Fartin A, Casas J. Orientation towards prey in antlions: efficient use of wave propagation in sand. *Journal of Experimental Biology* 210, 3337-3343 (2007).

- Finally, it seems rather difficult to get an idea of the venom composition of the two species. Based on Figure 3, the compositions don't seem all that different. It might have been interesting to show only the proteins confirmed by proteomics in a pie chart.

The venom protein composition may seem similar on first glance because the proteins in Figure 3 are grouped into very broad functional categories to reduce complexity. However, the categories are based only on the putative function and do not imply orthology. Moreover, a large number of identified proteins in *C. carnea* and *E. nostras* are uncharacterized and could not be grouped into functional categories. We found that particularly *E. nostras* venom contains protein families that do not occur in the *C. carnea* venom proteome. Many of them also lack homologs in the *C. carnea* genome. These results show that there are considerable differences in the venom protein composition that are not apparent from the general functional categories but may play important roles in the adaptation of *E. nostras* to its niche. As the

reviewer requested, we have added a supplementary pie chart (Figure S4) with the venom proteins confirmed by proteomics.

Figure S4: Venom proteins identified in the proteome of *Euroleon nostras* and *Chrysoperla carnea*. The identified proteins were grouped according to their protein family membership associations represented by color-coded blocks.

In the discussion, the authors discuss proteins that are unique to *E. nostras*, but the biological function of most of these proteins is unknown. Regarding PLA2, the authors attribute an offensive function to this toxin to paralyze prey, whereas in many studies PLA2 has a predominantly defensive function (e.g., it is present in exclusively defensive venoms such as that of honey bees). All these data do not allow us to conclude that the “dynamics of venom protein composition play an important role in the ecological diversification of neuropteran larvae”.

Although PLA2 mostly has a defensive function, it has also been identified as one of the main compounds in the offensive venom of predatory true bugs. We should therefore not exclude a toxic role for PLA2 in antlion venom. However, we understand that our conclusion may seem overstated and have therefore changed it as follows:

Lines 362-365: “Our results, particularly the number of species-restricted venom-related genes in *E. nostras* suggest that the antlion’s venom protein composition has evolved towards higher toxicity for prey insects than that of aphid lions, consistent with the differences in feeding ecology.”

II- In the M&M we learn that the authors have produced the genome of *E. nostras*. This is remarkable, but the genome seems to have very little use in the manuscript. It is not described at all in the results, nor does it seem to have been used to map the transcriptomic reads. Furthermore, the genomic data have not been deposited in a public database.

We used the genomes of *E. nostras* and *C. carnea* to identify species-restricted venom proteins and to obtain the full open reading frames of proteins selected for HCR-RNA-FISH (if the full sequence could not be retrieved from the transcriptome, as for *na2*). We have added a more detailed explanation in the results section.

Lines 174-176: “The full coding sequences of *pr-5*, *vsp* and *na1* were obtained from the *E. nostras* transcriptome, whereas the full coding sequence of *na2* was obtained from the *E. nostras* genome (Table S5).”

Moreover, we have deposited the genomic data in the Edmond data repository and have added the required information under “Data availability”:

Lines 603-605: “The genomic dataset for *E. nostras* has been deposited in the Edmond data repository and can be accessed under the following weblink: <https://doi.org/10.17617/3.YZVPXB>.”

Reviewer #2 (Remarks to the Author):

Review of COMMSBIO-24-2102-T

General comments

This manuscript describes a large body of work examining the production, composition, and activity of venoms produced by the Neuroptera (lacewings, antlions, and allies). As far as I am aware it is the first study to do so and as such it is a very important contribution to the field of arthropod venoms. There is a substantial amount of confusion in the literature to do with neuropteran venoms—some to do with the anatomy of venom production and some to do with supposed toxins produced by hypothesised bacterial symbionts. The objective approach of the current study clears away a lot of this confusion and that is very welcome.

I do however think the manuscript could be substantially improved. The most important things are to

clarify if it is known if the glands in the head (the maxillary-mandibular glands) could (considering the anatomy) or do contribute to the venom or if this is just unknown—I couldn't quite work this out from reading the current manuscript. As far as I can tell, the glands called the 'venom gland' and 'lateral gland' definitely do contribute to the venom, and if this is right I'd suggest they are called 'medial venom gland' and 'lateral venom gland' throughout. The 'maxillary-mandibular gland' name is currently quite confusing and it could be called something else e.g. 'cephalic gland'. I'd encourage the authors to discuss how each of these glands corresponds to glands or nomenclature identified/used by previous authors (include citations) but also introduce a sensible, useful nomenclature that will simplify future studies of neuropteran venoms.

We agree with the reviewer that the current nomenclature (venom gland, lateral gland, and maxillary-mandibular gland) used in the literature is confusing. We therefore welcome the proposal and have changed the terms used in the manuscript to medial venom gland, lateral venom gland, and cephalic gland, respectively. We explain the new nomenclature in the discussion as follows:

Lines 260-262: "To simplify the differentiation of these glands, we propose a new nomenclature, namely the medial venom gland, the lateral venom gland, and the cephalic gland (formerly the venom gland, lateral gland, and mandibular-maxillary gland, respectively)."

We understand that the contribution of the three different glands may be unclear. Because these glands are small and difficult to access, we were unable to dissect them for gland-specific RNA-Seq and instead performed RNA-Seq on different parts of the head. Genes encoding venom proteins identified in the proteome were expressed in all head tissues. Expression in the maxillae can clearly be attributed to the medial and lateral venom glands (supported by our HCR-RNA-FISH data). Considering the anatomy of the glandular systems resolved by micro-CT and histology, our results strongly suggest that the venom proteins expressed in the remaining head tissues are produced in the cephalic gland. Therefore, we are convinced that all three glands produce venom proteins. We have added the following sentences in the results and discussion sections, respectively, to clarify this:

Results (Lines 131-134): "Genes encoding proteins that were identified in the venom proteome were expressed in all head tissues, suggesting that the medial venom gland, the lateral venom gland and the cephalic gland are involved in venom production (Figure 3)."

Discussion (Lines 276-277): "This indicates that part of the venom is injected through the venom canal and part through the food canal."

One section of the manuscript examines the presence of bacteria, presumably because of the previous studies of supposed bacterial endosymbionts. The methods used here are kind of cool but the results are actually not of high relevance on reflection and I suggest the authors consider if it is really necessary or if it could be removed or transferred to supplementary material. Confusingly, one species had plenty of bacteria and one had none, but this prompts us to ask what the relevance of this outcome is. If we believe the result from the species that has no bacteria associated (and I'm not sure I do because it's hard to see what the positive control would be and I've never heard of an insect with no microbiome), it suggests the endosymbionts are not important for venom function. For the species with numerous associated bacteria, does the result suggest that bacterial endosymbionts are important for venom function? No. I think the proteomic results and the annotation of the proteins present in venom as observed by MS being of insect

origin are the important results from which it can be concluded that the previous studies on toxins produced by bacterial endosymbionts are not relevant to neuropteran biology. This comes with the caveat that to really make this result airtight, the MS data should be searched against databases including bacterial sequences and I don't think that was done.

The assumption that neuropteran toxins are produced by bacterial symbionts is persistent in the literature and is often referred to as a prime example of insect venoms produced by microbial symbionts. We believe it is very important to refute this and provide a solid basis for future research to avoid future misunderstandings. To do so, we want to support our results by incorporating different methods. Therefore, we do not agree that the FISH analysis lacks relevance for this study and would prefer to keep it in the main manuscript.

We understand that it is surprising that there seem to be no bacteria in any tissues of *E. nostras*. To ensure that this is reproducible, we repeated the experiments with two more specimens and came to the same outcome. In one specimen, we found traces of degenerated bacteria in the gut. We have now stated the number of replicates in the materials and methods section to clarify this (**Line 550**).

The fluorescent probes used in this experiment generally target eubacteria and are routinely used successfully in our laboratory, including for the FISH analysis of *C. carnea*. We can therefore rule out problems with the fluorescent probes. Furthermore, we can ascertain that the fixation and FISH procedure generally worked, because a clear DAPI signal can be seen on the images. We are therefore confident that the experiments worked and that there really are no living bacteria in *E. nostras*.

Although most insects have a microbiome, several different animal taxa including insects across multiple orders have been reported to carry few or even zero microbes (Hammer TJ et al. (2019) Not all animals need a microbiome. FEMS Microbiol Lett 366, fnz117). Antlion larvae have a unique gut morphology, in which the midgut is separated from the hindgut, thus preventing the excretion of feces at the larval stage. It is therefore possible that *E. nostras* larvae need to suppress the microbes in their gut to avoid colonization with pathogens, unlike other insects that can simply excrete unwanted microbes.

In conclusion, the absence of bacteria in *E. nostras* provides strong evidence that the venom is exclusively produced by the insect itself, contrasting with earlier reports for another antlion species. As the reviewer correctly points out, the presence of bacteria in *C. carnea* does not mean that they are involved in venom production, and we also state this in the discussion.

As the reviewer suggested, we have searched the MS data against the SwissProt database that contains bacterial proteins to further support our results. We detected traces of bacterial proteins in the *E. nostras* venom proteome, which are not considered reliable given the low number of peptides detected and the low intensity. However, we found few bacterial proteins with higher confidence in the *C. carnea* venom proteome, including bacterial chaperonins such as GroEL, which is one of the highest expressed proteins in many bacteria. Given the high numbers of bacteria detected in *C. carnea* tissues, it is therefore not surprising to find small amounts of GroEL in the venom of *C. carnea*. GroEL has also been suggested to be a major insecticidal toxin in the venom of the antlion *M. bore*. However, our insecticidal assays showed that *C. carnea* venom is not toxic against insects, thus strongly indicating that these proteins do not act as toxins in *C. carnea*. We added the following information in the results and discussion:

Results (Lines 123-125): “We detected traces of bacterial proteins in the venom proteome of *E. nostras*, which are not considered reliable given the low number of peptides detected and the low intensity (Table S2). *C. carnea* venom contained few bacterial proteins including bacterial chaperonins (Table S3).”

Discussion (309-320): “Moreover, we found that the venom system and digestive tract of *E. nostras* are devoid of extracellular and intracellular bacteria (Figure 5b, d) and the venom contains no bacterial proteins. In contrast, *C. carnea* carried intracellular bacteria in the medial and lateral venom glands and the cephalic gland (Figure 5f, Figure S11c). In consistence with this, *C. carnea* venom contained few bacterial proteins, including chaperonins such as GroEL (Table S3). Given that GroEL is one of the highest expressed proteins in obligate pathogens and symbionts and is particularly upregulated under stress conditions^{48,49}, it is not unexpected to find small amounts of it in the venom of *C. carnea*. It plays important roles in folding and unfolding of proteins but has also been suggested to be a major insecticidal toxin in the venom of the antlion *M. bore*^{29,49}. However, we could show that *C. carnea* venom has no significant paralytic or lethal activity against insects, thus strongly indicating that neither GroEL nor other bacterial proteins in the venom of *C. carnea* act as insecticidal toxins.”

- 29 Yoshida, N. *et al.* Chaperonin turned insect toxin. *Nature* **411**, 44-44 (2001).
 48 Wilcox, J. L., Dunbar, H. E., Wolfinger, R. D. & Moran, N. A. Consequences of reductive evolution for gene expression in an obligate endosymbiont. *Molecular microbiology* **48**, 1491-1500 (2003).
 49 Fares, M. A., Moya, A. & Barrio, E. GroEL and the maintenance of bacterial endosymbiosis. *TRENDS in Genetics* **20**, 413-416 (2004).

We have added two supplementary tables Table S2 and Table S3 showing the putative bacterial proteins in the venom proteomes of *E. nostras* and *C. carnea*, respectively.

Table S2: Putative bacterial proteins detected by LC-MS/MS in the venom proteome of *Euroleon nostras*.

Protein	Organism	Protein Description	Total Peptides	Unique Peptides	Total Intensity
sp Q24800 SEVE_ECHGR	Echinococcus granulosus	Severin	1	1	1.73E+08
sp B9JQW4 G6PI_AGRVS	Agrobacterium vitis	Glucose-6-phosphate isomerase	1	1	1.58E+08
sp A5CU93 KAD_CLAM3	Clavibacter michiganensis subsp. michiganensis	Adenylate kinase	1	1	9.65E+07
sp C0ZVT7 EFTU_RHOE4	Rhodococcus erythropolis	Elongation factor Tu	1	1	0
sp A6L8N5 ATPA_PARD8	Parabacteroides distasonis	ATP synthase subunit alpha	1	1	0

Table S3: Putative bacterial proteins detected by LC-MS/MS in the venom proteome of *Chrysoperla carnea*.

Protein	Organism	Protein Description	Total Peptides	Unique Peptides	Total Intensity
sp Q983S4 CH604_RHILO	Mesorhizobium japonicum	Chaperonin GroEL 4	9	1	7.73E+07
sp B9K1Y8 CH60_AGRVS	Agrobacterium vitis	Chaperonin GroEL	4	1	7.73E+07
sp P24753 G3P_SEROD	Serratia odorifera	Glyceraldehyde-3-phosphate dehydrogenase (Fragment)	3	1	6.71E+07
sp Q2NUI3 DPS_SODGM	Sodalis glossinidius	DNA protection during starvation protein	3	3	6.02E+07
sp Q98F85 PAL_RHILO	Mesorhizobium japonicum	Peptidoglycan-associated lipoprotein	4	3	5.22E+07
sp Q2NRD5 PGK_SODGM	Sodalis glossinidius	Phosphoglycerate kinase	3	1	2.55E+07
sp B1MGH7 EFTU_MYCA9	Mycobacteroides abscessus	Elongation factor Tu	2	1	1.02E+07
sp Q11HA6 EFTU_CHESB	Chelativorans sp.	Elongation factor Tu	2	1	7.00E+06
sp Q98GS0 IHFB_RHILO	Mesorhizobium japonicum	Integration host factor subunit beta	1	1	6.46E+06
sp Q8UGF0 RL9_AGRFC	Agrobacterium fabrum	50S ribosomal protein L9	2	1	6.07E+06
sp P23847 DPPA_ECOLI	Escherichia coli	Dipeptide-binding protein	1	1	3.29E+06
sp Q8ZB98 IPYR_YERPE	Yersinia pestis	Inorganic pyrophosphatase	1	1	3.15E+06
sp Q89AJ7 ODO1_BUCBP	Buchnera aphidicola subsp. Baizongia pistaciae	Oxoglutarate dehydrogenase	1	1	2.56E+06

sp Q0W1V5 RS4_METAR	Methanocella arvoryzae	30S ribosomal protein S4	1	1	2.32E+06
sp Q2NWS0 RL11_SODGM	Sodalis glossinidius	50S ribosomal protein L11	1	1	1.96E+06
sp P50204 PHAB_PARDE	Paracoccus denitrificans	Acetoacetyl-CoA reductase	1	1	1.72E+06
sp P29272 G3P_CERSP	Cereibacter sphaeroides	Glyceraldehyde-3-phosphate dehydrogenase	1	1	1.66E+06
sp P53573 ETFA_BRADU	Bradyrhizobium diazoefficiens	Electron transfer flavoprotein subunit alpha	1	1	1.62E+06
sp Q2NW23 IF2_SODGM	Sodalis glossinidius	Translation initiation factor IF-2	1	1	7.45E+05
sp Q1GVQ9 RS15_SPHAL	Sphingopyxis alaskensis	30S ribosomal protein S15	1	1	6.15E+05
sp Q49842 MIAB_MYCLE	Mycobacterium leprae	tRNA-2-methylthio-N(6)-dimethylallyl adenosine synthase	1	1	0.00E+00
sp Q2NRG1 SYK_SODGM	Sodalis glossinidius	Lysine--tRNA ligase	1	1	0
sp Q98DU8 SECB_RHILO	Mesorhizobium japonicum	Protein-export protein SecB	1	1	0
sp Q2S6J0 RS2_SALRD	Salinibacter ruber	30S ribosomal protein S2	1	1	0

Overall I recommend this important manuscript is accepted after revisions in response to the issues I have raised.

Specific issues

- Line 29: “The Neuroptera is” should be either “Neuroptera is” or “The Neuroptera are”

We have changed this to “The Neuroptera are...” (Line 31)

- Line 57: “There is a lack of molecular data to confirm such symbiotic relationships”. In the light of this study, this statement is kinder and more vague than is warranted. Consider adding a sentence to the effect of “However, these previous studies did not demonstrate the occurrence of these bacterial toxins in venom or demonstrate they have any relevance to natural prey capture by neuropterans”.

We have changed this to “However, there is a lack of molecular data to confirm the presence and ecological relevance of bacterial toxins in neuropteran venoms.” (Lines 57-59)

- Line 97: What is the rationale for calling one a venom gland and the other a lateral gland? If this nomenclature is based on previous studies please cite them.

The two maxillary glands were called the venom gland and lateral gland here based on previous studies. However, we have changed the nomenclature to lateral venom gland and medial venom gland as discussed in our earlier response.

- Line 119: “peptides ranging from < 10 to > 260 kDa in size” should be “peptides ranging from < 10 to proteins > 260 kDa in size”

We have changed this in the manuscript as recommended. (Line 123)

- Line 122: Please elaborate in more detail the ‘transcriptome analysis’ used to find these other 40% of putative toxins

The transcriptomic analysis is explained in detail in the materials and methods section. The putative toxins were selected as follows:

Lines 500-503: “Venom protein candidates were selected in the transcriptome according to venom-specific annotations, tissue-specific expression and the presence of a signal peptide for secretion. The preselected putative venom proteins were complemented with the venom proteome, and only candidates that were present in the proteome and/or had a signal peptide were retained.”

- Line 133 comment: There are several proteins shared with true bugs, who have similar requirements for extraoral digestion, including e.g. gelsolin which you can imagine contributing to EOD

We agree that there are several interesting proteins that are also used by other venomous animals such as true bugs and are worth investigating, including gelsolin.

- Line 139: Could you reword to clarify you mean these proteins do not have homologues in the *C. carnea* genome?

We have rephrased this as follows:

Lines 148-151: “Moreover, we found 34 proteins in the *E. nostras* venom proteome that have no homology to known proteins in public databases. We aligned the coding sequences of these 34 proteins with the *E. nostras* and *C. carnea* genomes and found that all of them were encoded in the *E. nostras* genome, whereas 28 of them had no homologues in the *C. carnea* genome (Table S4).”

- Line 162: Please indicate how or why you selected these genes/proteins for analysis.

We rephrased the next sentence to clarify how we selected these genes, as follows:

Lines 172-174: “From the genes that were most strongly expressed in the maxillae ($\log_2(\text{TPM}+1) > 13$), we selected two genes broadly expressed across all tissues (*pr-5*, *na1*) and two genes specifically expressed in the maxillae (*vsp*, *na2*) (Figure 4c).”

- Line 177: Again, if the lateral gland and venom gland are both contributing to the venom should you reconsider the nomenclature? Should they be lateral venom gland and medial venom gland?

We have changed the nomenclature as discussed in our earlier response.

- Line 178: I can’t find NA2 in the spreadsheet, not sure why? The manuscript says it was detected in venom but searching either the name or parts of the sequence shown in Figure S7 yield nothing.

The gene name *na2* was used for simplicity and corresponds to the contig name Enos_GHB_C9242 in the spreadsheet. We added a supplementary table (Table S5) with the simplified gene names and the corresponding contig names and coding sequences:

Table S5: Full coding sequences and corresponding contig names of the genes *pr-5*, *vsp*, *na1* and *na2* from *Euroleon nostras* that were used for hybridization chain reaction RNA fluorescence *in situ* hybridization.

Gene name	Contig name in the transcriptome	Full coding sequence
pr-5	Enos_GHB_C137	ATGACAACAATCAGTATTTAGTGCCTTTGTTATTTAAACGGTTTATGTTGTTAACGGTTCGA GAATTTCAATTTTTGAATAATTACGGACAACAATTATGGCTCGGAATACAAGGAAATAGTGG AAAAGGGACACCAATGGCGGTGGATTGTATTAATCTGGACAAGATCATCCATACAT GTTGCTGACGATTGGGGTGGTCTGTTTTGGGCAAGAACCAGTTGCAATGGCGGTAATAATC ATTGTGAAACAGGCGATTGTGGCAATCGTTTAGAGTGCAGGATTAATGGTGGTGCACCTCC CGTCAGTTTAGCTGAAATAACCTGAAAGGATGGGGAGGCCCTTGATTATTATGACCTTTTCAT TGGTCGATGTTTCAATATACCAATTGCTATGGAACCATTAGGTGGACAAGGTGATGGTAGT CAATATAGTTGTAAGAGCCACATGCCATGCAATATAAATGGTATTGTCCAATGAATT ACGTTTATGGTCGAACGGGAATGTAGTTGGATGTAATCAGCTTGTAGCATTCAATACGG ATCAATATTGTTCCGTGGTGCACATAATCGTCCAGAAACATGTCGGTCAAGTATTGGCCG GTTAATTATCCAGATGGTTAAAGATCGTTGTCCAGATGCATACAGTTACGCTTACGATGAT CATAAGAGTACATTTACATGTCGTGCACCAGCATATTTGGTGACCTTCGGTTAA
vsp	Enos_GHB_C52324	ATGGATCCGCGACTCACTAGTCTATTTGTATTATAATATTTTGTGGACAAATTACAGCA AATGAAAAATGTACAACCTGATGCTGATCCCGCGAATGTGTTGATGTGGACAATTGTCC ACAACGAAATCGGCACTGTATCGTATCTATGGCCAGGATGCCATACAACATATGGAACGTG ATGTGCACGTAACATGTCTACGTGTGGATGGAATGATGAACGTGATGTGCGGAAGGTGTG TTGTCCACAGAACTAGCGCGATATGGATACCGATGCTATGCAATGTTTGGCGAACCGG AAGGAACGTTTGAACAGCAATCCGGGAGTATGTGGTGTACGTACGTAATGATTCATAC TCACGTATTGTTGGCGGTTTTACTGTTAAAAAAGGACAATTTCCATGGATTGTTGCGTTGGG ATATGACGAAGGGAATTTACTATCCAGACTGGAATGTGGAGGTTCCCTAATTACACCAA ATCATGATTAACAGCTGCTCATTGTGCATTACCAAACTAATCAAAGTAAAAATTGGAACCC TGAATCCATGGGCACCTGGATATTTAGTCGGCGATGTTATCGAAATCCACAACACGAAAA TATGATAACGTGACGGATGTTTTCGATATTGCAATTTTGAATAATTCATGGAGAGATTATT GGCAACCAACAGAAATTAACCATCAGATGAAGCCCGACTAATCTGTTTCCGCAAGAAAT GGATATGCGCAAAAAATCGTATGAGGGACTTTCCATTTGGTGGTGGGCGTTATG GAATGGAATACAACAGACTAGCACCGATTTATTAGCTGTACAAGTAAAAATTGTTAAACA GAGTATTTGTCGTAAGAAATATGCAAAATTTCAAACGTATTAAGGTGACAAATAATGTAATGT GTGCCGATATAAACCGATCGTTTGGATTCGTGCAAGGTGATTCTGGTGGTCCATTAATG TTACCACATTTAGGTCCAGACGGTCTGATCTATTATTCAAATTTGGTATTGTGCGAAAGGT TATCGATGTCCTAAAGGGATGCCAGCCGTTTATACCGGTGTCGGTAACTTTATTGAATG GATTAATGCAAGATTAATTTATAG

na1	Enos_GHB_C1012	ATGTTCTACAACTTTTCTCATCTGTCTTATTGCTTTGATCGCTGTTGCCACAGCTGATCCAC AATTATACTACAGCGGATATTATCCAGGTGCCCGTGCCTATGTCTGGTGGCTATTACCCATATG CCGGTTACCCAGCCGTAGCTTATTATGGCAAGTGA
na2	Enos_GHB_C9242	ATGTCCAATTTTTTAAATTGTTGTTACAGTATTATGTGTTGTTGGTTTATACGAATGCAA TTGTGAATGACCTTAAAAATTCTTCATCGCTCATACCGGAACATAAGGAAGATAAAAAGGAT CAACAAGAACTCCTATAGTTTCACGAGGATCATTCCAAGACGATTATTACCAACATCGACA AATACAACATAAAACATACAAAATTTATCGATGCACGTATCCTGTTTCGACGTAGTCAACAGA AACAGTATCCTCACGATCATGGGCAGCACCCCTCGCGATCAGGAACAATACCCTTACGACAAA AGACAGTATCCTCACGATCAAGAGCAGCACCCCGTGATAAAGGGCAGTATCCTCGGGATC AGGAACAGTACCCCTATGACAAAAGGACAGTACCCTCATGATCAAGAACAGTATCCCCACGAT AAAGGGCAGCATCCTCGGGATCAGGAACAATACCCCTCACGATAAAGGACAGTACCCACAAG ATAAAGACAATACCCACATATAAAGGAAAATATCCGTATTCAGCATATATAGGTGATGAA AAAACACCAAATGAATTACCAAAAAGAAATCAATGGACTTCGTTAACTACCAATAATTGATTAT CGATTTTGCAAATTTATTGGTCCGTTACGGATTCTGATGATATGCCTCCATCTCCTATCCCTT CACCAGATCCATTACCAAGACAAGGACATGATTATCAACGTATGGATCGTGGCCGGTTCCA TCTGTGGACCATTCAGTTATCCAATTAGTGGTCATCGTCCACGAATACCTACTAGTTATATA CCATCATTCGGTCCGGATTTTTTTGTATAG

In the spreadsheet, we only included the translated ORFs obtained from the transcriptome assembly. The full sequence of *na2* was retrieved from the antlion genome. We have clarified this in the manuscript as follows:

Lines 174-176: “The full coding sequences of *pr-5*, *vsp* and *na1* were obtained from the *E. nostras* transcriptome, whereas the full coding sequence of *na2* was obtained from the *E. nostras* genome (Table S5).”

- Line 198: It’s weird that no bacteria at all were found in any tissues (including the exterior of the animal and the gut). Did these experiments just not work? A control here would have been helpful to interpret this experiment. But also, I don’t really find the results of this experiment that relevant. If there are bacteria there or not, what matters is what is in the venom, which you have shown is encoded by insect genetic material. I almost feel like this paper would be benefit from removal of this section on localisation of microbes and addition of a statement stating this finding explicitly (e.g. at the end of the section titled Protein composition of neuropteran venoms, you could state “These results demonstrate that neuropteran venom toxins are encoded by insect genetic material and not bacterial symbionts as previously hypothesised (cite papers).”

Please see our earlier response covering this issue.

- Line 221: 3 h seems like a pretty long timepoint for a venom that is meant to be used for prey capture to first show effect, could the authors comment on this?

The medial and lateral venom glands are small and difficult to access so it was not possible to dissect them or extract venom. This is why we had to rely on the use of head homogenates and venom collected using a prey dummy for the insecticidal assays. Both methods result in diluted venom, which is probably the reason for the delayed effects. When we feed antlions with caterpillar larvae, we usually observe paralysis within seconds to minutes (depending on the prey size).

- Line 244: I thought the stuff in the venom was from two different glands, the ‘venom gland’ and ‘lateral gland’ but not the maxillary-mandibular gland which opens into the food canal. But it says three here?

We assume that all three glands contribute to venom protein production and have clarified this in the manuscript, as described in our earlier response.

- Line 251: Should probably mention EOD in the Introduction somewhere

We agree with the reviewer and have mentioned EOD in the introduction:

Lines 34-36: “Their specialized piercing-sucking mouthparts form elongated pincers, which are used to catch and feed on prey using extraoral digestion (EOD).”

- Line 255: This suggests the maxillary-mandibular gland contributes to venom but I’m unsure what the evidence for this is. I think the way the Results is written and presented should be altered to make a few things more clear: (a) Do you expect the maxillary-mandibular gland to be in the head RNA-Seq sample but the venom gland and lateral gland in the maxillae RNA-Seq sample?

We assume that the maxillary-mandibular gland (now called cephalic gland) is included in the head RNA-Seq sample and the venom gland and lateral gland (medial venom gland and lateral venom gland, respectively) are included in the maxillae RNA-Seq sample. Based on the expression profiles, we therefore assume that all three glands contribute to venom protein production and we have clarified this point in the manuscript. Please see our earlier responses covering this issue for the further detailed justification of these assumptions.

(b) Can you do a statistical analysis to ask the question of if venom proteins (those detected by LC-MS/MS) are more highly expressed in the maxillae RNA-Seq sample than the head RNA-Seq sample. If yes, explicitly argue that this result suggests venom is made in the venom gland and lateral gland but not the maxillary-mandibular gland. Or if not, say that it means the maxillary-mandibular gland likely contributes to venom production.

We used pooled samples (with no biological replicates) for the RNA-Seq experiments so we cannot apply robust statistical analysis. However, Figure S5 shows that all head tissues express venom proteins detected in the venom proteome and that many of them are specifically expressed in the anterior head, indicating that they are produced by the maxillary-mandibular gland (now called cephalic gland).

(c) In light of the anatomy elucidated in this paper, does it make sense that the maxillary-mandibular gland contributes to venom production?

We are convinced that the maxillary-mandibular gland (now called cephalic gland) contributes to venom production. The underlying anatomy indicates that secretions from the medial and lateral venom gland are injected through the venom canal, whereas secretions from the cephalic gland are injected through the food canal. As stated above, we have added this explanation in the discussion of the revised manuscript.

- Line 325: In what sense are these genes orphans?

These genes are described as orphan genes because they currently have no (known) homologs even in closely related species such as *C. carnea*, which suggests they are species-restricted.

- Line 331: Why probably from non-coding RNA?

We propose that *na2* evolved from non-coding DNA because it shows no significant homology to bacterial or fungal genes, making acquisition by horizontal gene transfer unlikely. Moreover, we could not identify any homologs in other species and no paralogs in the antlion genome, which argues against *na2* having

evolved via a gene duplication event. The most parsimonious explanation would be the evolution from non-coding DNA.

- Line 399: Wouldn't you usually use data independent acquisition in this type of study?

We are confident that data independent acquisition is sufficient in combination with the very detailed tissue-dependent RNA-Seq analysis.

- Line 402: Would these methods have detected toxins produced by bacterial endosymbionts if present?

If not, should the MS files be searched against bacterial databases to check?

We have searched the MS data against a bacterial database as suggested (please see our earlier response covering this issue in detail).

- Line 410: Should you give a pH value for the amm bicarb?

We have added the pH value (8.5) for the ammonium bicarbonate.

- Line 463: Are the results from these BUSCO comparisons in the manuscript or supp data somewhere?

We have added the BUSCO comparisons in the supplement (Table S6).

Table S6: Results of the BUSCO analysis for the transcriptome assemblies of *Euroleon nostras* and *Chrysoperla carnea* and the genome assembly of *E. nostras*. C = complete BUSCOs, S = complete and single-copy BUSCOs, D = complete and duplicated BUSCOs, F = fragmented BUSCOs, M = missing BUSCOs.

Dataset	Species	BUSCO result
Transcriptome	Euroleon nostras	C:86.0%[S:83.6%,D:2.4%],F:7.0%,M:7.0%
	Chrysoperla carnea	C:89.3%[S:88.2%,D:1.1%],F:4.8%,M:5.9%
Genome	Euroleon nostras	C:98.6%[S:96.1%,D:2.5%],F:0.5%,M:0.9%

- Line 473: I think it would be better to only call those peptides and proteins that were detected in venom putative venom toxins (throughout the manuscript) but if the authors think differently that is fine.

We would like to keep the term “putative venom proteins” for all proteins, including those that were not identified in the proteome but feature a signal peptide for secretion.

- Line 475: I didn't realise this whole genome was assembled during the current study until now, maybe make this clearer with a sentence or two in the Results or elsewhere

We used the genomes of *E. nostras* and *C. carnea* to identify species-restricted venom proteins and to obtain the full ORFs from proteins selected for HCR-RNA-FISH (if the full sequence could not be retrieved from the transcriptome, as for *na2*). We have added a more detailed explanation on this in the results section.

Lines 149-151: “We aligned the coding sequences of these 34 proteins with the *E. nostras* and *C. carnea* genomes and found that all of them were encoded in the *E. nostras* genome, whereas 28 of them had no homologs in the *C. carnea* genome (Table S4).”

Lines 174-176: “The full coding sequences of *pr-5*, *vsp* and *na1* were obtained from the *E. nostras* transcriptome, whereas the full coding sequence of *na2* was obtained from the *E. nostras* genome (Table S5).”

- Line 566: The link isn't live yet, but does the transcriptomic data deposited include both the raw reads and assembled transcriptome?

We only deposited the raw reads.

- Has the genomic data also been deposited somewhere?

We have deposited the genomic data in the Edmond data repository and have added the required information under “Data availability”:

Lines 603-605: “The genomic dataset for *E. nostras* has been deposited in the Edmond data repository and can be accessed under the following weblink: <https://doi.org/10.17617/3.YZVPXB>.”

REVIEWERS' COMMENTS:

Reviewer #1 (Remarks to the Author):

I have no further comments on this new version.
I enjoyed reading this article. Great work!

Reviewer #2 (Remarks to the Author):

All issues raised in the previous review have been dealt with in a satisfactory manner in the revised manuscript and rebuttal document. I suggest this paper is ready for publication. The only further recommendation I have is that the assembled transcriptome, and the CDS and amino acid sequences of the detected putative toxins, be submitted to public databases if they have not already. I congratulate the authors on their interesting study!

Reviewer #1 (Remarks to the Author):

I have no further comments on this new version.
I enjoyed reading this article. Great work!

We thank the reviewer for the valuable suggestions and the positive feedback.

Reviewer #2 (Remarks to the Author):

All issues raised in the previous review have been dealt with in a satisfactory manner in the revised manuscript and rebuttal document. I suggest this paper is ready for publication. The only further recommendation I have is that the assembled transcriptome, and the CDS and amino acid sequences of the detected putative toxins, be submitted to public databases if they have not already. I congratulate the authors on their interesting study!

We thank the reviewer for the valuable suggestions and the positive feedback. We have uploaded the transcriptome assemblies as well as the contig and amino acid sequences of the putative venom proteins to the Edmond data repository.